# On Memory and Generalization in the Era of Linear Recurrence

## Abstract

Memory is crucial for the ability to store and retrieve prior knowledge when that information is gathered as a continuous stream that cannot be processed all at once. For decades, various types of artificial recurrent neural networks (RNNs) have been designed and improved to handle sequential data, incorporating memory in different ways. Transformers have become the most widely adopted architecture to deal with sequential data, while more recently structured state-space models (SSMs) and linear RNNs were put forward for their improved computational efficiency. While these families of models have been studied on various synthetic and real-world tasks, the generalization abilities of these newer models remain a topic of ongoing exploration. In particular, there is a gap in the current literature regarding the length generalization of models on sequence modeling tasks, both across models and across tasks. For models, while numerous studies have investigated the generalization of RNNs and Transformers to longer sequences, there is not much work devoted to such studies for SSMs or linear RNNs. Regarding tasks, one limitation of current works is their focus on formal language tasks for studying the generalization of sequence modeling. In contrast, the deep learning literature often introduces a variety of other tasks to assess the specific capabilities of deep learning models on sequential data. In this paper, we take a step toward addressing this gap by comparing the generalization abilities of all three families of algorithms across tasks that impose different memory requirements and are of special interest to the deep learning community, namely, copying tasks, state tracking tasks, and counting tasks. Our results show that despite their great efficiency, state space models seem to be less able than the nonlinear recurrent models to generalize to longer sequences.

## 1 Introduction

The ability to effectively learn from sequences forms a significant part of deep learning research given the wide amount of data in the real world that comes under this form. Various advancements have been made throughout the years by first determining limitations in existing methods, which led to developments in architecture and algorithms to overcome them. From the early recurrent networks, such as vanilla recurrent neural networks (RNNs) (Jordan, 1986; Rumelhart et al., 1986) which suffered from the *vanishing/exploding gradient* problem (Bengio et al., 1994; Pascanu et al., 2013), came variants that better overcame these through various modifications, such as adding gating components (Hochreiter & Schmidhuber, 1997; Cho et al., 2014; Chung et al., 2015), designing better initialization of recurrent weights (Le et al., 2015; Tallec & Ollivier, 2018; van der Westhuizen & Lasenby, 2018), parametrizing weights as orthogonal matrices (Henaff et al., 2016; Jing et al., 2019) and using non-saturating activation functions (Chandar et al., 2019). Then, the introduction of Transformers (Vaswani et al., 2017) addressed the issue of parallelizability and enabled the learning of dependencies of any length, thanks to the attention mechanism (Bahdanau et al., 2015) and led to a large shift towards this new architecture. However, due to the quadratic complexity of the attention mechanism with respect to the sequence length, the sequence length that Transformers can process is practically limited. Finally, state space models (SSMs) (Gu et al., 2021; 2022b) were developed with the claim of solving these remaining issues through the specific initialization of parameters of the recurrence matrix (Gu et al., 2020;

2022a), leading to an increased surge in interest in linear recurrence (Orvieto et al., 2023; Qin et al., 2023; De et al., 2024; Beck et al., 2024) under the belief that such a paradigm can avoid prior limitations.

In an attempt to understand the use of memory in sequence modeling, Deletang et al. (2023) consider a range of recurrent neural networks with various ways of incorporating memory, as well as Transformers, and compare their performance on a set of formal language tasks that differ in their memory requirements in the context of the automata theory. As a method of evaluating whether the trained RNN has learned the correct algorithm to solve the corresponding task, they analyze the extrapolation capability of the trained models by looking at their generalization to longer sequences than those seen during training. Interestingly, they find remarkable similarities between the behavior of the RNN models and their corresponding automata.

In their work, as well as in related research on the generalization of RNNs, such as (Wang & Niepert, 2019), the tasks considered are those typically examined in the context of automata theory and formal languages. In this paper, we are concerned with the tasks that are of particular interest and relevance to neural network studies, namely, copying memory, state tracking, and counting. Some of these tasks have emerged from the interest in modeling complex sequential data with deep neural networks, with all being crucial for benchmarking and evaluating different deep learning algorithms. Our interest in these tasks stems from their representation of significant challenges in sequential models. The copying memory task (Hochreiter & Schmidhuber, 1997; Arjovsky et al., 2016) is a benchmark extensively used in the literature to assess how sequential models address the vanishing gradient problem and handle long sequence modeling. Moreover, this task requires the memorization of a sequence of data, making it an ideal test for neural networks with robust memory structures. While many variants of RNNs have successfully solved the copying task, few studies test whether they learn the correct algorithm and hence, generalize. On the other hand, certain state tracking tasks may be too complex in terms of circuit complexity for linear sequential models like structured state space models (S4) to solve (Merrill et al., 2024). Therefore, various versions of these tasks with varying levels of difficulty are used to benchmark linear sequential models, such as S4-types, against nonlinear ones. Regarding generalization performance, while earlier works such as (Deletang et al., 2023) have studied the capability of Transformers and sequential networks on state tracking tasks, their sequential models are limited to nonlinear RNNs. Only a few recent studies have investigated the ability of certain linear sequential models to generalize to longer sequences Sarrof et al. (2024); Grazzi et al. (2024). We will discuss the relevant part of their results in Section 4. Exploring the generalization capabilities of a wider range of linear sequential models and tasks can provide valuable insights into the performance of these models compared to nonlinear ones. Therefore, aside from the gap in task types, another aspect of the generalization studies in learning models that requires further exploration is the variety of models that have been examined. Our work aims to address these gaps by employing a similar approach to the one used by Deletang et al. (2023). Finally, the counting task, in the context of timing (Gers & Schmidhuber, 2000; Gers et al., 2002), demonstrates the ability of sequential models to measure the temporal distance between events in sequences with rhythmic patterns. This capability is crucial for tasks such as music processing.

In the following sections, after reviewing some related works, we define a set of representative tasks, each necessitating different types of memory usage to be solved effectively. Next, we consider a set of commonly used neural network models with different ways that memory is utilized for sequence modeling and evaluate these architectures and tasks under various settings to understand how memory is employed in learning from sequences. In summary, our contributions are as follows:

1. We define a memory taxonomy that applies to the deep learning synthetic tasks studied while relating them and their memory requirements to corresponding tasks in formal language theory.

2. We empirically examine whether and how memory use in different neural architectures affects learning and generalization across tasks of varying difficulty.

3. Following the claim of Merrill et al. (2024) that SSMs, such as S4 and Mamba, and Transformers share a similar inability to learn hard state tracking tasks due to their parallelizable nature, we empirically demonstrate that, even in generalization, these state space models exhibit behavior similar to Transformers on regular language tasks (Deletang et al., 2023). Specifically, even when successfully trained on solvable state tracking tasks, SSMs fail to generalize to longer sequences.

Overall, our work contributes to this growing field by extending the studies of memory and generalization to include more recent sequential models within the current state of deep learning research.

## 2 Related Work

**Memory in Neural Networks** In sequence modeling, recurrent neural networks stood out due to their ability to learn and carry out complicated transformations of data over extended periods of time, as well as the potential to simulate arbitrary procedures with proper construction (Siegelmann & Sontag, 1995). They handle variable-length sequences by having a recurrent hidden state whose activation at each time is dependent on that of the previous time through a general update equation

$$\boldsymbol{h}_t = \boldsymbol{f}(\boldsymbol{W}\boldsymbol{h}_{t-1} + \boldsymbol{U}\boldsymbol{x}_t) \tag{1}$$

where $\boldsymbol{h}_t$ and $\boldsymbol{x}_t$ are the hidden state and input vector at time $t$, $\boldsymbol{W}$ and $\boldsymbol{U}$ are the weight matrices parameterizing the RNN, and $\boldsymbol{f}$ is an activation function of choice. However, due to the gradient scaling problems, the hidden state acts only as a short-term memory (Miller, 1956) and is limited in the duration and quantity of the stored information. Attempts to alleviate this issue have primarily consisted along the direction of architectural design, such as the long short-term memory (Hochreiter & Schmidhuber, 1997) and gated recurrent units (Cho et al., 2014). Later, in an attempt to integrate better memory capabilities in sequential models, explicit memory modules (Graves et al., 2014; Gulcehre et al., 2017; 2018; Chandar et al., 2019) have been successfully explored to extend the capabilities of sequential models. This class of models, known as memory-augmented neural networks (MANNs), utilizes external memory, usually a matrix, to store information. A controller within the network learns to both read from and write to this external memory. This paradigm provides the model with greater flexibility to retrieve past information, forget, or store new information in the memory. Yet, these also pose challenges. In particular the additional complexity of managing an external memory structure and designing efficient algorithms for memory interactions, leaving it underexplored how memory explicitly helps with sequence modeling tasks.

**Memory and Extrapolation to Unseen Examples** In addition to helping RNNs overcome gradient scaling issues and achieve faster convergence on sequence modeling tasks (Graves et al., 2014; Chandar et al., 2019), memory cells have been shown in the literature to significantly improve the generalization of memory-augmented models to unseen samples, such as samples with sequence lengths longer than those seen during training. Graves et al. (2014) first showed that their Neural Turing Machine (NTM) with an LSTM controller not only converges faster to a lower minimum than LSTM for the memory-intense task of copying memory but also that in contrast to LSTM, the trained NTM can perform copying on sequences of more than twice as long as the ones seen during training. Wang & Niepert (2019) then speculated that the poor extrapolation capabilities of RNNs stem from a lack of regularization in their hidden space, leading to the memorization of data rather than accurately storing the sequence state at each step. They propose a state regularization technique with a state extraction method for the automaton corresponding to the trained RNN and empirically show the benefit of this process for the generalization to out-of-distribution (OOD) examples on tasks requiring some levels of memorization, such as balanced parenthesis (BP) task. One implicit assumption here is that the memory cells are required for the model to extrapolate, and state regularization optimizes the model's use of both its hidden state and its memory. Therefore, they consider the LSTM model for that task. The intuition comes from the fact that the balanced parenthesis task is a context-free language in the context of formal language theory, requiring a push-down automaton to express arbitrary depths of nested parentheses. Their experiment reveals that regularization prompts the LSTM to use its memory cell, rather than the hidden state, to track the nesting depth, akin to how the push-down automaton uses its stack. Consequently, a regularized RNN with a memorization mechanism becomes particularly advantageous for this task.

Finally, Deletang et al. (2023) conducted an exhaustive empirical study on the relationship between the architecture of sequence models and their ability to generalize to longer sequences. Their study spans a wide range of tasks and the corresponding state-of-the-art architectures. They show that while for higher levels of the hierarchy, different types of RNNs encounter difficulties in extrapolating beyond certain lengths, all of them, including the basic RNN and LSTM, generalize to a significant degree to longer lengths on regular

language tasks such as parity and modular sum. This confirms that vanilla RNNs are capable of simulating finite state automata.

**Counting with Neural Networks**   Counting is a trivial yet crucial skill possessed by the human mind and can be split into three distinct groups: item counting in arrays, arithmetic, and event sequencing (Noël, 2009; Logie & Baddeley, 1987). The greatest utility of counting for real-world applications is for tasks that require timing and sequence modeling, such as queuing, contextual awareness, and time series predictions. In the past, it was observed that recurrent models such as LSTMs that were extended by 'peephole connections', allowing them to observe their internal states, were able to time and count various target signals (Gers & Schmidhuber, 2000). More recently, it was shown that the popular Transformer models are also able to count, however, they are limited by the dimension of the model, and when performing counts on the most frequent element, single-layer Transformers are unable to learn this objective (Yehudai et al., 2024).

In the next section, we categorize the memory requirements for solving the deep learning tasks considered in this work and highlight their relationship to the categorization of related tasks in formal language theory.

## 3   Task-Based Memory Taxonomy

One well-established approach to formalizing memory classification in machine learning is through the Chomsky hierarchy, which hierarchizes formal language tasks based on their complexities and the type of automata that can solve them. For instance, finite automata correspond to regular language tasks with limited memory requirements, whereas pushdown automata, with their stack-based memory, can handle more complex context-free languages. By drawing analogies between neural networks and different types of automata, we can comment on the networks' capabilities in performing specific tasks and, therefore, their expressivity (Weiss et al., 2018; Deletang et al., 2023). However, it is important to note that there is no one-to-one correspondence between neural networks and automata. A key reason is that automata have discrete states, while the hidden states of RNNs exist within a continuous space. Therefore, empirical testing is necessary to validate the analogies and conclusions drawn about neural network capabilities[1]. In line with this argument, in this section, we review various levels of memory required for different learning tasks. We discuss the actual tasks considered in this study and their memory requirements in order to build a better taxonomy to distinguish our analysis. Notice that the categorization of memory in learning tasks presented here is based on our interpretation, drawing from both machine learning and neuroscience literature.

**Stateful Memory**   We define the first type of memory requirement for sequences as *stateful* memory tasks. In this setting, it is possible to predict the next token based solely on the current *internal state* and the current input token. That is, if the internal state is a sufficient statistic for the history, then the problem is Markovian. Therefore, the primary learning problem in these scenarios becomes learning such statistics. Stateful memory is the type of memory required for problems where an internal state with a suitable constant size is enough to learn such statistics for a sequence regardless of its length.

Such tasks are called regular languages in formal language theory and are recognized by a finite state automaton[2]. Stateful memory tasks include state tracking tasks such as bit parity tracking, where the model is given a sequence of 0s and 1s and must determine the current number of 1s in the sequence, either 0 or 1 (even or odd parity). In this paper, experiments on stateful memory tasks are conducted on algebraic groups of different computational complexities, such as $\mathbb{Z}_{60}$, $\mathbb{A}_4 \times \mathbb{Z}_5$, and $\mathbb{A}_5$, which are studied by Merrill et al. (2024). Solving these tasks requires understanding the current state, which is either one of $k$ position on a circle in the case of $\mathbb{Z}_k$ or permutation of $k$ numbers in the case of $\mathbb{A}_k$, and combining it with the current input token, which is a number of steps to move or an order to permute the system, in order to predict the correct output token (a new position or permutation). Merrill & Sabharwal (2023) suggest that any architecture that can parallelize computation over a sequence inherently lacks the ability to represent

---

[1]Aside from the well-known works on extracting automata from trained RNNs, Wang & Niepert (2019) show how to extract a finite set of automata states being learned by RNNs during training.

[2]One subtle difference is that in formal language theory, most tasks are defined in the context of acceptance and rejection, while the tasks of our interest are of the transduction type. The way we relate these two types of problems is similar to Deletang et al. (2023).

languages of a specific complexity[3] (problems that are $NC^1$-hard), including $\mathbb{A}_5$. As such, we are motivated to consider these tasks as examples of scenarios that only require stateful memory, since the output is always deterministically decided based on the previous state and incoming input, and can still be difficult to solve due to circuit complexity considerations.

State tracking tasks are representative of model capabilities in narrative understanding, such as discourse understanding or entity tracking (Kim & Schuster, 2023) and hence are significant in benchmarking different sequential models in the current state of language model research.

**Stable Memory**   While stateful memory tasks do not require keeping a single element in memory for a very long time, this is the case of what we define as *stable* memory tasks. These tasks could depend on a single token, but that token has to be kept in memory for a potentially long period before the information is used for the output. While they may not require learning a complex internal state representation as the input token could just be copied, they do require learning what must be stored in memory and keeping it in memory in order to learn the task. These tasks are distinct from stateful tasks in two key ways. Firstly, by definition, they necessitate a memory mechanism capable of retaining information for a long time. Secondly, the state size might need to expand depending on the amount of information that needs to be stored. This second characteristic implies that a constant-size state is inadequate for stable memory problems *with arbitrary sequence lengths*. Consequently, the formal language counterparts for these tasks reside higher in the Chomsky hierarchy than regular languages, including context-free and context-sensitive languages, which require additional memory structures such as stack and tape in their respective automata. However, note that in the specific cases where the sequence lengths are upper-bounded, the problem can be solved by a finite state automaton.

Examples of tasks demanding stable memory are the copying (Hochreiter & Schmidhuber, 1997; Arjovsky et al., 2016) and denoising (Jing et al., 2019) tasks. In the copying task, the model is given a sequence of $S$ tokens from a predefined vocabulary, followed by a sequence of $T$ noise tokens. After that, a specific indicator token tells the model to reproduce the initial $S$ tokens from the beginning of the sequence. An example is shown as:

$$\underbrace{\text{X} \quad \text{Y} \quad \text{Z}}_{\text{Sequence to copy of length } S} \quad \underbrace{\boxed{\text{B}} \quad \boxed{\text{B}} \quad \boxed{\text{B}}}_{T \text{ Blank tokens}} \quad \underbrace{\boxed{\text{C}}}_{\text{Indicator to start copying}} \quad \underbrace{\text{X} \quad \text{Y} \quad \text{Z}}_{\text{Target Sequence}}$$

The denoising task, also called selective copying, is similar to the copying task, except that the noise tokens are scattered in between the elements of the sequence of random tokens to be copied, still resulting in a sequence of $S + T$ tokens). An example is shown as:

$$\underbrace{\text{X} \quad \text{Y} \quad \boxed{\text{B}} \quad \boxed{\text{B}} \quad \text{Z} \quad \boxed{\text{B}}}_{S + T \text{ Interspersed Tokens}} \quad \underbrace{\boxed{\text{C}}}_{\text{Copying indicator}} \quad \underbrace{\text{X} \quad \text{Y} \quad \text{Z}}_{\text{Target Sequence}}$$

These tasks have been widely used to evaluate the capacity of models to learn long-range dependencies in sequences. Learning the denoising task is considered more challenging because the model must learn a more complex filtering/ignoring mechanism, as shown by Gu & Dao (2024). While those blanks may initially overfill the network memory in both tasks, they clearly indicate the end of the sequence to memorize in the copying task but not in the denoising task. In both cases, the memory of the non-blank tokens must be stable over the processing of the blank tokens. That is, the activation should not decay nor be altered by blank inputs. The gradient must also freely flow through time to learn the task correctly. If the memory is stable, the network should generalize to longer sequences of blanks.

**Counting Memory**   Another critical role of memory is to perceive duration in order to learn to produce some output at a specific moment in time. This requires an understanding of counting or timing. We define this as *counting memory*. This type of memory resides between the stateful and stable ones previously defined. While counting and timing tasks require noting time alongside learning the model's state, the required memory is simpler than the stable memory: instead of storing some specific content, the number of certain items or time steps need to be tracked and stored. The counterparts in formal languages are counter

---

[3]We use the term language as defined in formal language terminology.

languages which, aside from regular languages, include some context-free and context-sensitive languages. If the duration is bounded, then a finite automaton can solve the task. Otherwise, even a pushdown automaton is insufficient as what is needed is the stack size, not its content. Like state tracking, the internal state must change while the network is counting, and like copying, the gradient must flow from the timed-output token back from the initial counting trigger.

A simple yet representative task can be derived from classical and temporal conditioning (Gallistel & Gibbon, 2000) where models are autoregressively trained with *on/off* (or 1/0) signals of varying lengths. The objective is to learn the duration of the *on* signal, which is fixed, to accurately predict when it will turn *off*. This duration, known as the interstimulus interval (ISI) in the conditioning literature, is interleaved with a random interval of *off* tokens, called the intertrial interval (ITI). The network that successfully learns to count should focus on learning the ISI duration, as its end is fully predictable, whereas the beginning of the ISI is impossible to predict precisely. A crucial aspect of this timing task is that the actual task (timing the ISI) can start anywhere in the sequence, and the end to be predicted depends entirely on stable counting or time tracking from the ISI onset. The memory must then be fully reset to handle the next ISI in the sequence. In animals, the task does not get more difficult or easier as long as the ITI/ISI ratio remains constant (Gallistel & Gibbon, 2000). However, for neural networks, a longer ITI should increase the training difficulties (Rivest et al., 2010).

## 4 Experiments

In this section, we empirically evaluate the performance of different types of sequential models on the tasks described in Section 3. Given the nature of the tasks discussed, it follows that the varying inductive biases inherent to different neural network architectures can enable some to outperform others. In Appendix A, we describe the architectures we use in our experiments: RNN (Rumelhart et al., 1986), LSTM (Hochreiter & Schmidhuber, 1997), GRU (Cho et al., 2014), NRU (Chandar et al., 2019), Transformer (Vaswani et al., 2017), S4D (Gu et al., 2022a), and Mamba (Gu & Dao, 2024), and briefly discuss their memory properties. Additional implementation details are provided in Appendices B, D and H. We assess each model's ability to both learn the tasks and generalize to OOD examples as a means of probing whether the models have learned the correct algorithm. In our state tracking, copying, and denoising experiments, the OOD examples are sequences longer than the training ones. In counting, it is the ITI and ISI that are longer.

Among the key findings, only vanilla and gated RNN can generalize to longer test sequences in the state tracking task, while single-layer Mamba and S4D could not even learn it. Also, none of the multi-layer SSM models could generalize to longer sequences after having learned the task. In copying and denoising, only gated RNNs and Mamba can generalize to longer blank intervals, but none can generalize well to longer patterns to memorize. Finally, in counting, while Mamba, LSTM, and GRU had trouble generalizing and properly resetting their counter, vanilla RNN and S4D learned the task and passed both generalization tests, which suggests the high sensitivity of the task to overfitting. Transformers were unable to generalize on any of the tasks.

| Task | $\mathbb{Z}_{60}$ | | | | $\mathbb{A}_4 \times \mathbb{Z}_5$ | | | | $\mathbb{A}_5$ | | | |
|---|---|---|---|---|---|---|---|---|---|---|---|---|
| **Model** | RNN | LSTM | GRU | NRU | RNN | LSTM | GRU | NRU | RNN | LSTM | GRU | NRU |
| **Length** 128 | ✔ | ✔ | ✔ | ✔ | ✔ | ✔ | ✔ | ✔ | ✔ | ✔ | ✔ | ✔ |
| 256 | ✔ | ✔ | ✔ | ✔ | ✘ | ✔ | ✘ | ✔ | ✘ | ✔ | ✘ | ✘ |
| 512 | ✘ | ✔ | ✔ | ✔ | ✘ | ✔ | ✘ | ✘ | ✘ | ✘ | ✘ | ✘ |

Table 1: An extension of Merrill et al. (2024)'s on longer length state tracking tasks. (✔) indicates that a single-layer model attains $> 90\%$ accuracy on a held-out validation set (averaged over 5 seeds), (✘) means otherwise.

| Task | $\mathbb{Z}_{60}$ | | | | | | | | | $\mathbb{A}_4 \times \mathbb{Z}_5$ | | | | | | | | | $\mathbb{A}_5$ | | | | | | | | |
|---|---|---|---|---|---|---|---|---|---|---|---|---|---|---|---|---|---|---|---|---|---|---|---|---|---|---|---|
| **Test Length** | 1X | 2X | 3X | 4X | 5X | 64 | 128 | 256 | 512 | 1X | 2X | 3X | 4X | 5X | 64 | 128 | 256 | 512 | 1X | 2X | 3X | 4X | 5X | 64 | 128 | 256 | 512 |
| Mamba | ✔ | ✗ | ✗ | ✗ | ✗ | ✗ | ✗ | ✗ | ✗ | ✔ | ✗ | ✗ | ✗ | ✗ | ✗ | ✗ | ✗ | ✗ | ✔ | ✗ | ✗ | ✗ | ✗ | ✗ | ✗ | ✗ | ✗ |
| Transformer | ✔ | ✗ | ✗ | ✗ | ✗ | ✗ | ✗ | ✗ | ✗ | ✔ | ✗ | ✗ | ✗ | ✗ | ✗ | ✗ | ✗ | ✗ | ✔ | ✗ | ✗ | ✗ | ✗ | ✗ | ✗ | ✗ | ✗ |
| S4 | ✔ | ✗ | ✗ | ✗ | ✗ | ✗ | ✗ | ✗ | ✗ | ✔ | ✗ | ✗ | ✗ | ✗ | ✗ | ✗ | ✗ | ✗ | ✔ | ✗ | ✗ | ✗ | ✗ | ✗ | ✗ | ✗ | ✗ |
| IDS4 | ✔ | ✗ | ✗ | ✗ | ✗ | ✗ | ✗ | ✗ | ✗ | ✔ | ✗ | ✗ | ✗ | ✗ | ✗ | ✗ | ✗ | ✗ | ✔ | ✗ | ✗ | ✗ | ✗ | ✗ | ✗ | ✗ | ✗ |
| RNN | ✔ | ✔ | ✔ | ✔ | ✔ | ✗ | ✗ | ✗ | ✗ | ✔ | ✔ | ✔ | ✔ | ✔ | ✗ | ✗ | ✗ | ✗ | ✔ | ✔ | ✔ | ✔ | ✔ | ✗ | ✗ | ✗ | ✗ |
| LSTM | ✔ | ✔ | ✔ | ✔ | ✔ | ✔ | ✗ | ✗ | ✗ | ✔ | ✔ | ✔ | ✔ | ✔ | ✔ | ✔ | ✔ | ✔ | ✔ | ✔ | ✔ | ✔ | ✔ | ✗ | ✗ | ✗ | ✗ |
| GRU | ✔ | ✔ | ✔ | ✔ | ✔ | ✔ | ✗ | ✗ | ✗ | ✔ | ✔ | ✔ | ✔ | ✔ | ✔ | ✔ | ✔ | ✗ | ✔ | ✔ | ✔ | ✔ | ✗ | ✗ | ✗ | ✗ | ✗ |
| NRU | ✔ | ✔ | ✔ | ✔ | ✔ | ✔ | ✗ | ✗ | ✗ | ✔ | ✔ | ✔ | ✔ | ✔ | ✔ | ✔ | ✗ | ✗ | ✔ | ✔ | ✔ | ✔ | ✗ | ✗ | ✗ | ✗ | ✗ |

Table 2: Evaluating the ability of models to extrapolate from their training length to longer sequences. Here, we provide an example where $k = 8$. The number of layers of each model depends on the ability to learn the training length (`1X`). (✔) indicates that the model can attain $> 90\%$ accuracy on a held-out validation set (averaged over 5 seeds), and (✗) means otherwise. We evaluate on extrapolation up to 5 times the training length and fixed lengths of 64, 128, 256, and 512.

## 4.1 State Tracking

State tracking experiments use the same setting as in (Merrill et al., 2024), specifically, they are based on three group operations: $\mathbb{Z}_{60}$, $\mathbb{A}_4 \times \mathbb{Z}_5$, and $\mathbb{A}_5$. In all our experiments, we control each layer to have approximately 3 million parameters. Explicit hyperparameters and details on training and generalization metrics are provided in Appendix B. The task performance is measured by the model's accuracy across the full sequence. The model is considered to have predicted a sample correctly if it can predict all tokens exactly within the sequence. Since all samples have the same length, overall performance on the task is measured by the number of correctly predicted samples within a held-out test set. As for generalization, a model is considered to have 'extrapolated' to a length only if it simultaneously achieves the predefined threshold accuracy on that length as well as on all shorter test lengths.

**Initial Long Sequence Results**  We begin by asking why learning group multiplication in $\mathbb{A}_5$ poses challenges for models. While one-layer models can handle sequences of length $k = 2$, some struggle as $k$ increases, unless model depth scales accordingly. Since the underlying operations are deterministic, they should ideally be learnable by a transition function dependent on state and input. Our central question regarding training performance is thus whether there is a length limit to the effectiveness of one-layer models. To test this, we increase $k$ beyond 25, exceeding the range explored by Merrill et al. (2024). We also study IDS4, a model they introduced, which claims to solve $\mathbb{A}_5$ tasks via input-dependent $\boldsymbol{A}$ transition matrices.

Table 1 shows that even recurrent models with nonlinearities, which are claimed to have the ability to model arbitrarily long state tracking problems, have limits on all tasks being tested. Note that other models (Transformer, Mamba, and S4D) cannot properly learn sequences with a single layer and therefore are excluded. IDS4, on the other hand, demonstrates significant numerical issues, which render it incapable of modeling sequences longer than 20 tokens, and hence is also excluded. A comparison with other recurrent models further shows that those with explicit memory structures (LSTM and NRU) are better able to learn sequences of increasing length. More details on the training dynamics of the nonlinear recurrent models can be found in Figure 4.

**Short-to-Long Extrapolation**  A particular feature of state tracking tasks is the deterministic nature of transitions, which means if a model properly learns the underlying transitions of a specific task, then it can trivially extrapolate to sequences of any arbitrary length, as the rules of the transitions do not change regardless of the length. Therefore, we explore how model performance generalizes to longer sequences.

Table 2 shows extrapolation performance from training on sequences of length 8. Interestingly, only nonlinear recurrent networks are capable of extrapolating to any degree past the length of sequences used for training; this suggests that parallelizable models fail to properly learn the task. Meanwhile, we observe that nonlinear recurrent models with explicit memory structures demonstrate improved extrapolation performance.

**Summary of Results**

1. All models have limits, both at training on long sequences and generalizing to longer sequences. In particular, despite theoretical arguments in (Merrill et al., 2024) about single-layer SSMs being able to solve easy state tracking tasks, such as $\mathbb{Z}_{60}$ modular sum, in our experiments, only nonlinear models can solve those tasks with one layer for long sequences.

2. The solution of finite-layer SSM models on the easy state tracking tasks does not extrapolate to longer sequences, suggesting that the model has not learned the correct algorithm. The results for S4D and Mamba are fully consistent with the theoretical and empirical analysis presented in (Grazzi et al., 2024), which attributes their lack of generalization to specific properties of their transition matrices: S4D's time-invariance and Mamba's nonnegativity, both of which hinder the ability of SSMs to accurately track states. In contrast, the outcome for IDS4 contradicts our initial expectations, as its transition matrix is both input-dependent and complex. We therefore hypothesize that, in this case, the limitation may arise from learning dynamics. These points are discussed in greater detail in Section 5.3.

## 4.2 Copying and Denoising

As defined earlier, the input of the copying and denoising tasks consists of $S$ random tokens, $T$ blank tokens, and an indicator token. In practice, the input has a length of $T + 2S$, where we append another $S$ blank tokens to the end of the input sequence, giving space for the model to produce the output tokens.

For training dynamics, we study the evolution of accuracy over the random sequence segment of length $S$. We have two experimental setups: one with all models having a similar number of parameters, and another with models having the same hidden size. For both cases, we consider both 1-layer and 2-layer architectures.

To study generalization to longer sequences, we consider two settings. In the first one, models are trained on sequences with fixed pattern length (10) and varying blank interval lengths (up to 50); in the second, pattern length varies (up to 10) while the blank length is fixed at 50. We evaluate extrapolation in both cases: the first setup tests on longer blank intervals (up to 500) with fixed pattern length 10, and the second on longer pattern lengths (up to 100) with fixed blank length (50). Intuitively, generalizing to memorize more tokens is expected to be more challenging because, from the perspective of formal language theory, it examines the ability of the model to handle the context-sensitive aspect of the task.

Finally, while some models do not learn the task within the maximum number of training steps (150K), we still keep them in our generalization analysis. The list of model hyperparameters is provided in Appendix D.

**Training on Mixed Blank Interval Length and Extrapolation to Longer Blank Interval Length**
Table 3(row 1) shows the results for training on mixed blank interval lengths but fixed pattern lengths for models with one and two layers, where different architectures have comparable sizes. Contrary to vanilla RNN, all gated RNN models successfully solve the task. While Mamba solves the task only with 2 layers, S4D exhibits the fastest convergence even with 1 layer (see also Figure 5 in Appendix E). The Transformer's slow convergence is primarily due to the lack of Positional Encoding (NoPE). Although a Transformer with sine-cosine encoding learns the task very quickly, that type of positional encoding significantly reduces its extrapolation capability and hence we did not include it in our results[4]. From Figure 9, we observe similar results for the case where all models have the same hidden size.

Figure 1 (a) shows the results of length generalization. Gated RNNs maintain strong performance on sequences with extended blank intervals. Interestingly, Mamba exhibits similar behavior, aligning with gated RNNs. In contrast, S4D, despite rapidly learning the task, struggles to extrapolate, indicating a tendency for its architecture to overfit. Figure 10 in Appendix F shows similar results on the generalization performance of architectures with the same hidden size.

**Training on Mixed Pattern Length and Extrapolation to Longer Pattern Length** Table 3 (row 2) shows the results for training on mixed pattern lengths but fixed interval lengths for models with one and two layers, where different architectures have comparable sizes.

---

[4]Deletang et al. (2023) study several other positional encodings including RoPE and ALiBi. We did not include them because Transformers were not the main focus of our work, and none of the encodings studied in their work resulted in a good generalization performance on the duplicate string task, which is a formal language task related to our copying task.

| Model | RNN | | LSTM | | GRU | | NRU | | TF | | S4D | | Mamba | |
|---|---|---|---|---|---|---|---|---|---|---|---|---|---|---|
| Layer | 1 | 2 | 1 | 2 | 1 | 2 | 1 | 2 | 1 | 2 | 1 | 2 | 1 | 2 |
| Copying $S = 10\ T \leq 50$ | ✗ | ✗ | ✔ | ✔ | ✔ | ✔ | ✔ | ✔ | ✗ | ✗ | ✔ | ✔ | ✗ | ✔ |
| Copying $S \leq 10\ T = 50$ | ✗ | ✗ | ✔ | ✔ | ✔ | ✔ | ✔ | ✔ | ✗ | ✗ | ✔ | ✔ | ✗ | ✔ |
| Denoising $S = 10\ T \leq 50$ | ✗ | ✔ | ✔ | ✔ | ✔ | ✔ | ✔ | ✔ | ✗ | ✗ | ✗ | ✗ | ✗ | ✔ |
| Denoising $S \leq 10\ T = 50$ | ✗ | ✔ | ✔ | ✔ | ✔ | ✔ | ✔ | ✔ | ✗ | ✗ | ✗ | ✗ | ✗ | ✔ |

Table 3: Evaluating the ability of single and dual layer models to learn the copying and denoising tasks on sequences of length $S = 10$ and $T = 50$ with a similar number of parameters within 150K time steps (averaged over 5 seeds). (✔) indicates that the model was able to learn the signal and (✗) indicates otherwise.

Figure 1 (c) shows the results of length generalization for different models with 1 and 2 layers where all models have the same number of parameters. The main observation here is that all models fail to extrapolate for pattern length twice the maximum training length and beyond. Figure 12 in Appendix F shows the results for the setting with models having the same hidden size.

**Denoising** Table 3 (rows 3-4) shows training results for the denoising task. That task is definitely more challenging for S4D and Mamba. Figure 7 and Figure 8 in Appendix E show the learning dynamics of the models with the same number of parameters on the denoising task for sequences of mixed blank interval lengths and mixed pattern lengths, respectively. The corresponding extrapolation results are shown in Figure 1 (b) and Figure 1 (d). Learning dynamics for models of the same hidden size are illustrated in Figures 13 and 15, and the results for their extrapolation are shown in Figures 14 and 16.

**Summary of Results**

1. All models struggle to generalize to sequences with longer pattern lengths. That is, they do not perform well when the amount of data they need to retain in memory increases. This happens while, in principle, the models have enough capacity to store the information of those long sequences. This result aligns with the observations in (Deletang et al., 2023), regarding the performance of various sequential models on the string duplication task from the context-sensitive class of the Chomsky hierarchy. They conclude that for this memory-intensive task, only Tape-RNN could extrapolate to longer strings, while other RNN models, including LSTM and GRU, failed to generalize beyond their training sequence lengths.

2. The matrix-valued hidden state of Mamba does not improve its extrapolation on the copying task compared to recurrent networks with vector-valued hidden state, implying that this mechanism of memory augmentation is still not helpful with promoting the model on the Chomsky hierarchy.

3. Specific to the denoising task, the superior performance of Mamba compared to S4D could be attributed to its input-dependent weights, which enhance the filtering ability required for the denoising task. In contrast, S4D is time-invariant and is expected to struggle on this task compared to the simple copying task.

## 4.3 Learning to Count

For the first task, we present models with a fixed-length stimulus (the ISI) and random ITIs. The signal is *on* for ISI intervals of $\{10, 30, 50\}$ consecutive time steps depending on the sequence length. The signal is *off* for ITI intervals randomly picked from intervals $\{[20, 40], [60, 120], [100, 200]\}$ depending on the sequence length. The length of the signals varies among 200 (short), 600 (medium), and 1000 (long) time steps. The signal is constructed by alternating ITIs and ISIs, where it always starts with an ITI followed by an ISI and this pattern repeats for the whole sequence length. Therefore, this memory task involves 2 components: the ability to reset at the onset of the stimulus and the ability to either count the duration of the ISI or to compute an offset time stamp to predict the ISI offset.

We evaluate models with two hidden dimensions 8 and 64. However, our results suggest that the task is trivial for the hidden size 8.

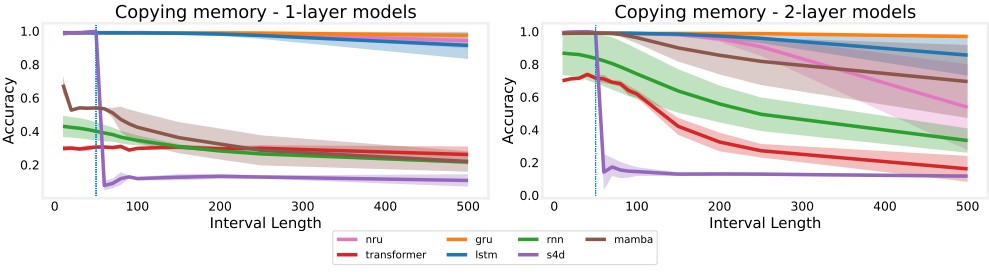

(a) Extrapolation to longer blank intervals for copying task.

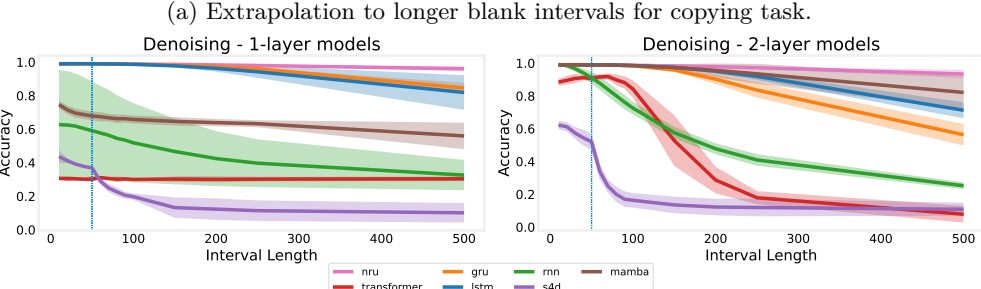

(b) Extrapolation to longer blank intervals for denoising task.

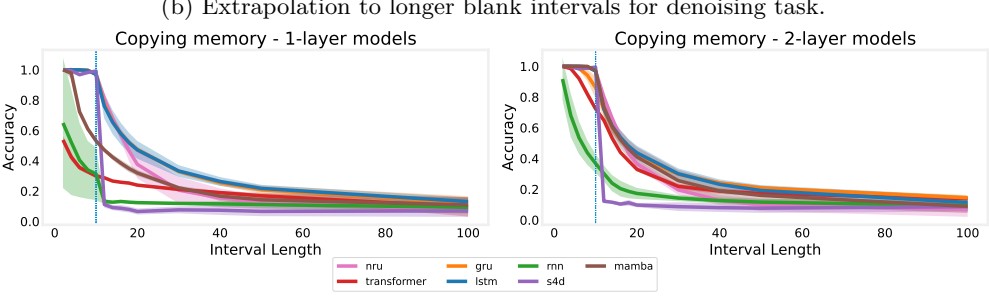

(c) Extrapolation to longer patterns for copying task.

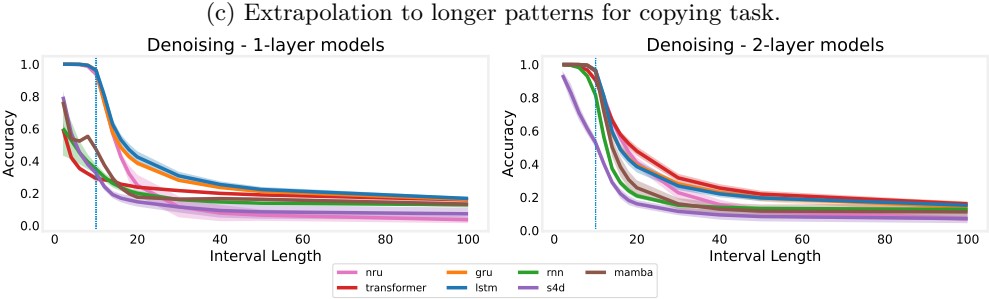

(d) Extrapolation to longer patterns for denoising task.

Figure 1: Extrapolation of (left) 1- and (right) 2-layer models with a comparable number of parameters trained on a mixture of shorter blank intervals (top 2 rows) and patterns with shorter lengths (bottom 2 rows). The dashed vertical blue line is the training range. Results are averaged over 5 seeds.

**Predicting Signal Offset** In Table 4, we report the ability of single-layer models to predict the desired offset for short, medium, and long ISIs. We consider a model successful if it predicts the ISI offset at least 3 out of 5 trials within the specific configuration.

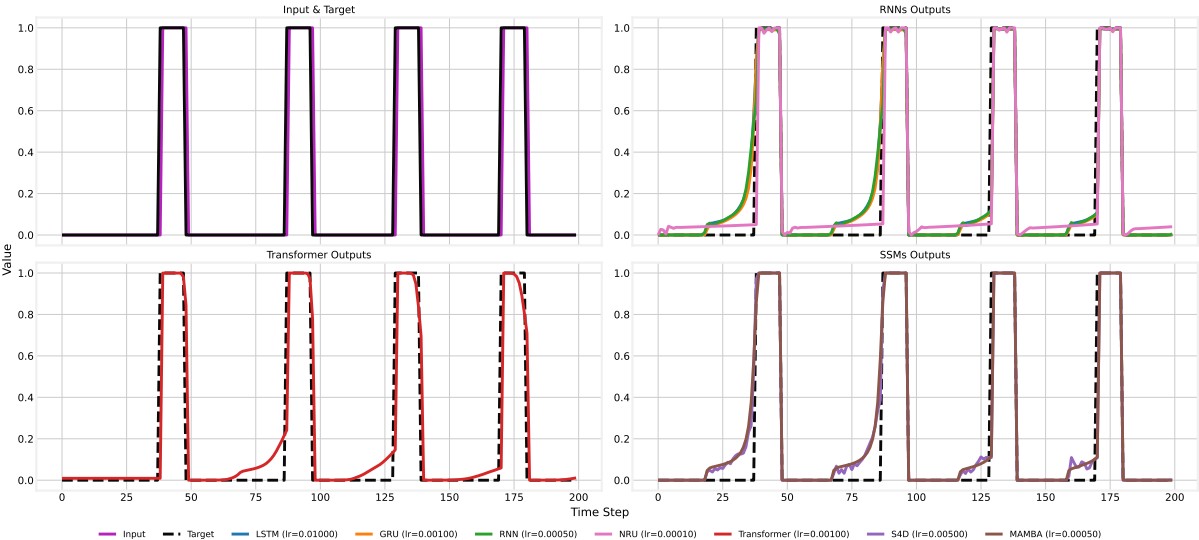

Figure 2: Model outputs for models with hidden size of 8, trained on 1000 epochs and a 200 sequence length. The Transformer and Mamba are two-layer models. All recurrent models have slight anticipation for the onset and correctly predict the offset. The Transformer model has anticipation for the onset but misses the desired offset (lower left panel). S4D and Mamba behave similarly to the recurrent models.

The most notable observation is that Mamba and Transformer fail on this task, while all RNN-type models solve it as expected. Interestingly, S4D also solves this task, while it is generally believed to be less expressive than Mamba. In Appendix I, we provide a solution that exists within the function class that can be modeled by S4D, but is out of reach for Mamba. This difference stems from the fact that the transition matrix of S4D is complex, while for Mamba this matrix is real. This is the point brought up in (Grazzi et al., 2024) about the inability of Mamba to solve the modular counting task, which is different from the counting task we study here but also has crucial similarities. Whether or not the S4D is learning the analytical solution with complex values that we suggest in Appendix I requires an analysis of the trained model that we postpone to future work. Also, it is interesting to investigate whether or not there is any analytical solution that Mamba can find, i.e., a real set of model parameters that can still solve the task. To further test Mamba and Transformer empirically, we trained a 2-layer version. While the 2-layer Mamba solved the task, the Transformer still failed on it, which we assume is due to the lack of positional encoding in our model.

**Phase Alignment Tests**   A model's ability to generalize can be inferred by the ability to phase align, defined as resetting its memory after seeing a stimulus. Here, we conduct two-phase alignment tests on single-layer models trained on short signals. The results are similar for medium and long sequences.

| Model | RNN | | LSTM | | GRU | | NRU | | TF | | S4D | | Mamba | |
|---|---|---|---|---|---|---|---|---|---|---|---|---|---|---|
| **Hidden Size** | 8 | 64 | 8 | 64 | 8 | 64 | 8 | 64 | 8 | 64 | 8 | 64 | 8 | 64 |
| **Length** 200 | ✔ | ✔ | ✔ | ✔ | ✔ | ✔ | ✔ | ✔ | ✘ | ✘ | ✔ | ✔ | ✘ | ✘ |
| 600 | ✔ | ✔ | ✔ | ✔ | ✔ | ✔ | ✔ | ✔ | ✘ | ✘ | ✔ | ✔ | ✘ | ✘ |
| 1000 | ✔ | ✔ | ✔ | ✔ | ✔ | ✔ | ✔ | ✘ | ✘ | ✘ | ✔ | ✔ | ✘ | ✘ |

Table 4: Performance of single-layer models on the offset prediction task after 1000 epochs, with hidden sizes 8 and 64, on all 200, 600, and 1000 time step sequences. (✔) indicates that the model was able to learn the signal and (✘) indicates otherwise. Results are reported based on five different seeds.

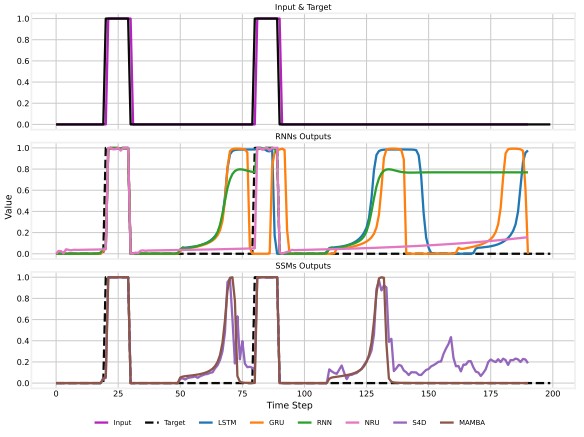 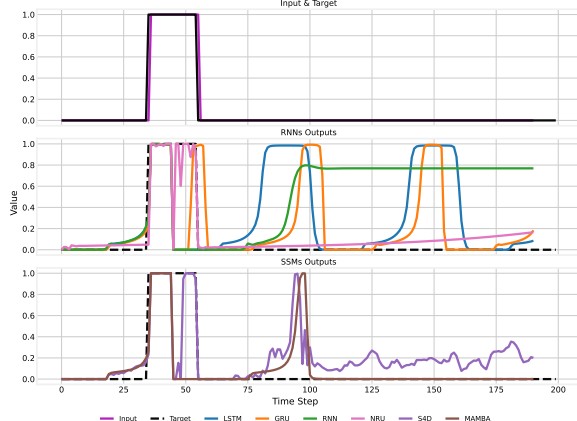

(a) Consecutive ISIs with intermediate ITI          (b) Double ISIs with temporal offset

Figure 3: Phase alignment test results for two ISI configurations: (a) Consecutive ISIs with an intermediate ITI and (b) Double ISIs with a temporal offset. All models other than Transformer, LSTM, NRU, and GRU demonstrate offset prediction in at least one condition, suggesting sensitivity to ISI/ITI structure. Note: Mamba models use two layers.

First, we evaluate whether models can reset their memory. To test this, a second ISI is placed after an initial ISI and a long ITI (longer than the ITI seen during training) at test time (Figure 3 (a)). This assesses the ability to consistently recognize the onset of a signal and begin counting. The desired behavior is thus for the model to correctly predict the end of the ITI and re-activate upon the appearance of the second ISI.

The second test evaluates if models can correctly reactivate their counting mechanism when encountering consecutive ISIs (Figure 3 (b)). After completing the first ISI, the model should deactivate briefly, as learned during training, then reactivate and continue counting for the second ISI's duration. This time, LSTM is the only recurrent model that fails to generalize for both hidden sizes.

Table 5 shows that among all recurrent networks, GRU (for Two ISI) and LSTM (for Double ISI) do not generalize for any of the hidden sizes. Trying out the generalization ability of 2-layer Mamba, we observe that it passes the consecutive ISI test for both 8 and 64 hidden unit configurations (Figure 3). We hypothesize that the failure of GRU and LSTM models on generalization is due to overfitting because vanilla RNN is already generalizing on both tasks and hence this cannot be about the lack of expressivity. Even for S4D, which is a linear model, we could find an analytic solution and empirically showed that it both solves the task and generalizes in both experiments.

**Summary of Results**

1. S4D, while being linear and time-invariant, solves the task and generalizes, suggesting that it has learned the correct automata in Figure 17. In Appendix I, we also suggested one possible solution that resides in

| | Model | RNN | | LSTM | | GRU | | NRU | | TF | | S4D | | Mamba | |
|---|---|---|---|---|---|---|---|---|---|---|---|---|---|---|---|
| | **Hidden Size** | 8 | 64 | 8 | 64 | 8 | 64 | 8 | 64 | 8 | 64 | 8 | 64 | 8 | 64 |
| **Test** | Two ISI | ✔ | ✔ | ✘ | ✔ | ✘ | ✘ | ✘ | ✔ | ✘ | ✘ | ✔ | ✔ | ✘ | ✘ |
| | Double ISI | ✔ | ✘ | ✘ | ✘ | ✔ | ✘ | ✔ | ✔ | ✘ | ✘ | ✔ | ✔ | ✘ | ✘ |

Table 5: Phase alignment results for single layer models on varying lengths on short signals sequences trained for 1000 epochs. (✔) indicates that the model was able to phase align at least 3 out of 5 times with the specific configuration, and (✘) indicates otherwise.

the function class that S4D models, but have not verified whether this is the solution found by gradient descent in our experiments.

2. Vanilla RNN also both solves the task and generalizes, which is expected from the fact that there exists a FSA (Figure 17), that simulates this task and that any FSA can be simulated by a vanilla RNN. Based on this, the occasional failures of GRU and LSTM suggest that this task is too sensitive to overfitting.

3. Contrary to S4D, Mamba, even with two layers, did not succeed to generalize in one of our designed phase alignment experiments. There are two possible explanations for this. First, this can be due to the training dynamics; similar to LSTM and GRU, Mamba can, in principle, solve the task but overfits and hence not generalize. Secondly, it can be an expressivity issue due to it lacking complex eigenvalues. This lack of complex eigenvalues was shown in (Grazzi et al., 2024) to prevent Mamba from solving a similar counting task, i.e., modular counting. That is, while S4D is believed to be less expressive than Mamba, it can solve modular counting because it is parameterized in the complex space.

4. Transformer fails to learn the task, regardless of the size. This aligns with prior work showing that single-layer Transformers struggle with implicit counting and temporal alignment (Yehudai et al., 2024).

## 5 Discussion

In this section, we provide an additional overview of the motivation behind our research, our interest in the topic of neural network memory, and its relevance to our generalization studies. After highlighting the gaps in the current literature that our work addresses, we review the main results of our experiments and the insights that our findings offer for future research.

### 5.1 Relating Memory and Extrapolation

Understanding deep learning models is crucial because these methods often operate as black boxes, making it difficult to interpret their solutions. It is essential to have insights and methods to interpret the model's solution to ensure that a model has learned the correct algorithm for a given problem. Extrapolation to OOD samples serves as a significant indicator of whether the model has internalized the correct algorithm. On the other hand, in the classical machine learning framework of formal language theory, there is a correspondence between different formal languages and various types of automata that differ in terms of their memory component. If an automaton with a finite number of states possesses the correct memory component for a task with given memory requirements, then in principle it should be able to learn the correct algorithm to solve the task and extrapolate to unseen examples. For example, a pushdown automaton generalizes on the task of balanced parentheses to longer open parenthesis depth than seen during training, while a finite-state automaton cannot realize this generalization. While there is no direct equivalence between neural networks and automata, and many tasks of interest in deep learning research may not directly translate to formal language tasks, valuable insights can still be drawn. Similar arguments can be applied to analyze the suitability of deep learning models for different tasks. Therefore, categorizing the memory mechanisms of these models and studying their generalization capabilities across various problems is crucial. This approach provides essential insights into developing better models and designing more informative benchmarking tasks.

### 5.2 Limitations of Prior Studies

Research on the computational power of neural networks in relation to automata has been conducted for various recurrent neural networks across different formal language tasks (Weiss et al., 2018; Wang & Niepert, 2019; Deletang et al., 2023). However, with the recent development of alternative novel architectures, more complete studies are still largely absent from the current literature. In terms of OOD generalization, some of these new architectures have demonstrated exceptional extrapolation capabilities to longer sequences for specific tasks, such as associative recall and induction heads (Gu & Dao, 2024; De et al., 2024). On the other hand, various works have identified failure modes in other tasks, particularly different types of state-tracking (Sarrof et al., 2024; Grazzi et al., 2024). Since each synthetic task represents different deep learning capabilities, there's significant potential to further improve these models. Addressing these gaps and optimizing for various task requirements could lead to more versatile neural networks.

### 5.3 Our Results and Future Directions

We presented our key findings on the models' learning and generalization abilities in the experiments section, organized by task. Here, we interpret the broader insights that emerge from the experiments collectively.

Our findings on nonlinear recurrent networks align with existing literature. These models exhibit an intrinsic ability to generalize on state tracking and counting tasks but struggle with copying and denoising, tasks previously shown (Deletang et al., 2023) to demand more sophisticated mechanisms, such as a tape structure for handling longer sequences.

The failure of Transformer models across all tasks is consistent with prior studies on similar challenges. Especially, parity as a state tracking task (Liu et al., 2023) and duplicate-string resembling the copying task (Deletang et al., 2023). For parity, beyond the hypothesis that positional encoding leads to out-of-distribution activations on longer sequences, another explanation concerns the expressive power of Transformers. As demonstrated by Liu et al. (2023), these models often rely on shortcut solutions that perform well within the training distribution but fail to generalize to unseen data.

Linear RNNs, or state-space models, represent the least studied case and the central focus of our work. Although existing studies are limited, our results are consistent with the current literature and offer additional insights.

At first glance, the inability of SSMs to extrapolate to longer sequences on solvable state tracking tasks may appear surprising in light of (Merrill et al., 2024). However, this behavior finds explanation in the later results of Grazzi et al. (2024) and Sarrof et al. (2024). As shown in their investigation into the expressivity of current SSMs on parity and more complex solvable state tracking tasks, specific design choices in the transition matrix hinder the models' ability to learn the correct algorithm and generalize to out-of-distribution samples. To solve these tasks, the transition matrix must be both input-dependent and possess complex eigenvalues. Current SSM implementations in the literature typically lack one or both of these properties. In our experiments, for instance, S4D is time-invariant, while Mamba's transition matrix is constrained to real eigenvalues.

That being said, a novel insight from our work emerges from experiments with IDS4, an input-dependent extension of S4 (Gu et al., 2022b) proposed by (Merrill et al., 2024) to solve state tracking tasks using a single layer. IDS4 represents a strong candidate for complex-valued, input-dependent SSMs and is therefore expected, in theory, to perform well on solvable state tracking tasks. Yet, its failure to generalize to longer sequences suggests that, even when intrinsic expressivity issues are addressed, training dynamics remain a limiting factor. This motivates a deeper analysis beyond the expressivity frameworks offered by (Sarrof et al., 2024; Grazzi et al., 2024), particularly into whether gradient-descent-based optimization can reliably discover correct solutions. Furthermore, if IDS4's failure to generalize stems from overfitting, this poses an important question for further research. The motivation for studying model performance on state tracking tasks lies in their relevance to real-world applications, such as code analysis and math problems, where tracking state is essential. In practice, large language models are employed for these tasks and are known to benefit from over-parameterization. While prior work demonstrates enhanced performance of language models, those modified to solve synthetic state tracking tasks, on math and code datasets (Grazzi et al., 2024; Siems et al., 2025), the impact of over-parameterization on state tracking remains an important avenue for deeper investigation.

Finally, although some argue that the concept of state, and thus the identification of automaton states in SSMs, is less well-defined than in RNNs (Merrill et al., 2024), numerous aspects of SSMs remain under active investigation, including the interpretation of their internal states. Given that state extraction offers one of the most direct and transparent ways to interpret a model's learning process, this represents a particularly promising direction for future study.

With such methods devised, it becomes possible to apply state regularization techniques developed for RNNs, enhancing the utilization of their hidden states (Wang & Niepert, 2019). If similar methods could be designed for SSMs, architectures with improved memory abilities like xLSTM (Beck et al., 2024) would greatly benefit from them. Implementing these strategies for SSMs could potentially unlock new levels of performance and efficiency in state management.

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

# A Models

In this appendix, we outline various models employed in our experiments in Section 4. These models are categorized into three types: recurrent neural networks, Transformers, and state space models.

## A.1 Recurrent Neural Networks

### A.1.1 Vanilla Recurrent Neural Networks

Recurrent Neural Networks (RNNs) are the basic architectures that model sequential data by having a recurrent hidden state whose activation at each time is dependent on that of the previous time, through a general update equation

$$\boldsymbol{h}_t = \sigma(\boldsymbol{W}[\boldsymbol{h}_{t-1}, \boldsymbol{x}_t] + \boldsymbol{b}) \tag{2}$$

with $\boldsymbol{W}$ being the weight matrix that parameterizes the RNN and $\sigma$ the activation function of choice (often chosen to be a tanh or sigmoid function). Note that our notation here is different from equation 1. $[\boldsymbol{h}_{t-1}, \boldsymbol{x}_t]$ means the concatenation of the hidden state and the input, and accordingly, the weight matrix $\boldsymbol{W}$ includes both $\boldsymbol{W}$ and $\boldsymbol{U}$ weight matrices of equation 1. We use the same notation in the subsequent sections for LSTM and GRU models as well.

Regarding the memory structure, RNNs only have a nonlinear recurrent activation, meaning that only their hidden state can be considered a memory module.

### A.1.2 Long Short-Term Memory

A major challenge with vanilla RNNs is the vanishing/exploding gradient problem which is due to the gradients becoming exponentially small or large during backpropagation. This results in long-term dependencies being hard to learn and hence has popularized variants which attempt to alleviate this concern. Long Short-Term Memory (LSTM) models are a popular variant of recurrent neural network that address this issue by extending the conventional RNN with their famous memory cell, denoted as $\boldsymbol{c}_t$ in the formulas below, along with input, forget, and output gates. These gates regulate the flow of information between the memory cell, the hidden state, and the output, determining what should be preserved or discarded. Specifically, the input gate ($\boldsymbol{i}_t$) controls the information added to the cell state ($\boldsymbol{c}_t$), the forget gate ($\boldsymbol{f}_t$) decides what information to discard, and the output gate $\boldsymbol{o}_t$, finally determines what the model outputs at each time step.

$$\begin{aligned}
\boldsymbol{f}_t &= \sigma(\boldsymbol{W_f}[\boldsymbol{h}_{t-1}, \boldsymbol{x}_t] + \boldsymbol{b_f}) & \boldsymbol{i}_t &= \sigma(\boldsymbol{W_i}[\boldsymbol{h}_{t-1}, \boldsymbol{x}_t] + \boldsymbol{b_i}) \\
\tilde{\boldsymbol{c}}_t &= \tanh(\boldsymbol{W_c}[\boldsymbol{h}_{t-1}, \boldsymbol{x}_t] + \boldsymbol{b_c}) & \boldsymbol{c}_t &= \boldsymbol{f}_t \odot \boldsymbol{c}_{t-1} + \boldsymbol{i}_t \odot \tilde{\boldsymbol{c}}_t \\
\boldsymbol{o}_t &= \sigma(\boldsymbol{W_o}[\boldsymbol{h}_{t-1}, \boldsymbol{x}_t] + \boldsymbol{b_o}) & \boldsymbol{h}_t &= \boldsymbol{o}_t \odot \tanh(\boldsymbol{c}_t)
\end{aligned} \tag{3}$$

Here $\boldsymbol{W_f}$, $\boldsymbol{W_i}$, $\boldsymbol{W_c}$, and $\boldsymbol{W_o}$ are the weight matrices associated with the forget, input, cell, and output state, respectively. Similarly $\boldsymbol{b_f}$, $\boldsymbol{b_i}$, $\boldsymbol{b_o}$, and $\boldsymbol{b_c}$ are the bias terms and $\boldsymbol{h}_{t-1}$ is the hidden state from the previous time-step. A sigmoid $\sigma$ nonlinearity is applied to the input, forget, and output gate and a tanh nonlinearity is applied to the candidate cell state $\tilde{\boldsymbol{c}}_t$. The $\odot$ is the element-wise product.

### A.1.3 Gated Recurrent Units

Gated recurrent units (GRU) are another form of recurrent model proposed to deal with the gradient scaling problem faced by conventional RNNs. Like LSTMs, GRU models make use of gates to update their hidden states; however, they have a much simpler architecture and no memory cells. A standard GRU consists of two gates in contrast to the three of an LSTM, a reset gate, $\boldsymbol{r}_t$, and an update gate $\boldsymbol{z}_t$. The reset gate considers the current input and determines how much of the hidden state, $\boldsymbol{h}_t$, should be used for the candidate state $\tilde{\boldsymbol{h}}_t$. The update gate determines what information should be forgotten from the previous hidden state, in other words, the extent to which the candidate cell should influence the current hidden state.

$$\begin{aligned}
\boldsymbol{r}_t &= \sigma(\boldsymbol{W_r}[\boldsymbol{x}_t, \boldsymbol{h}_{t-1}] + \boldsymbol{b_r}) & \boldsymbol{z}_t &= \sigma(\boldsymbol{W_z}[\boldsymbol{x}_t, \boldsymbol{h}_{t-1}] + \boldsymbol{b_z}) \\
\tilde{\boldsymbol{h}}_t &= \tanh(\boldsymbol{W_h}[\boldsymbol{x}_t, (\boldsymbol{r}_t \odot \boldsymbol{h}_{t-1})] + \boldsymbol{b_h}) & \boldsymbol{h}_t &= \boldsymbol{z}_t \odot \boldsymbol{h}_{t-1} + (1 - \boldsymbol{z}_t) \odot \tilde{\boldsymbol{h}}_t
\end{aligned} \tag{4}$$

Here, $\boldsymbol{W_r}$, $\boldsymbol{W_z}$, and $\boldsymbol{W_h}$ are the weight matrices associated with the reset gate, update gate, and candidate hidden state, respectively. Similarly, $\boldsymbol{b_r}$, $\boldsymbol{b_z}$, and $\boldsymbol{b_h}$ are the corresponding bias terms. $\sigma$ is again the sigmoid activation function applied to the reset and update gates, and tanh is applied to the candidate hidden state $\tilde{\boldsymbol{h}}_t$.

For LSTM and GRU, the memory mechanism includes the introduction of the gates and, hence, the additional control over what to keep in or remove from the hidden state. This results in a better flow of information over sequence steps and, hence, more effective memory. Also, compared with GRU, LSTM has an additional memory cell that gives it even more memorization capability; while the boundedness of the activations of the hidden states in LSTM and GRU limits their ability to deal with varying information along the sequence, the memory cell of LSTM provides an additional pathway for the information flow that mitigates this effect to some degree and hence is considered to provide it with a longer memory compared with GRU. Additionally, as pointed out in earlier sections, this separate cell state has been shown (Wang & Niepert, 2019) to result in a better allocation of memorization and state transition in learning tasks where we need both of these, i.e., both memorizing some elements in the sequence and recognizing the state of the sequence at the current step; a noteworthy example is the task of balanced parenthesis (BP) for which Wang & Niepert (2019) show how a neural network with distinct hidden state and memory cell, like LSTM, can be advantageous over an architecture where all structures for storing information are merged into a single hidden state, such as RNN and GRU.

### A.1.4 Non-saturating Recurrent Unit

The Non-saturating Recurrent Unit (NRU) (Chandar et al., 2019) is a memory-augmented recurrent neural network that was introduced mainly to further alleviate the problem of vanishing gradients in sequential models. To achieve this, NRU introduces two main modifications to the vanilla RNN: first, it has a memory vector that interacts with the hidden state; secondly, all the nonlinearities, including those in the recurrence equation as well as the ones used in the memory vector updates are ReLU (while keeping the introduction of nonlinearity optional); this further solves the problem of vanishing gradients in (gated) recurrent neural networks which is to some extent due to the saturation of gate or recurrent state values with vanishing derivatives. This led to better performance on several memory-intensive tasks, including copying and denoising, than its contemporary state-of-the-art recurrent architectures. The update equations for the state ($\boldsymbol{h}$) and memory ($\boldsymbol{m}$) are as below

$$
\begin{aligned}
\boldsymbol{h}_t &= \boldsymbol{f}(\boldsymbol{W_h}[\boldsymbol{x}_t, \boldsymbol{h}_{t-1}, \boldsymbol{m}_{t-1}] + \boldsymbol{b_h}) \\
\boldsymbol{\alpha}_t &= \boldsymbol{f}(\boldsymbol{W_\alpha}[\boldsymbol{x}_t, \boldsymbol{h}_{t-1}, \boldsymbol{m}_{t-1}] + \boldsymbol{b_\alpha}) \\
\boldsymbol{\beta}_t &= \boldsymbol{f}(\boldsymbol{W_\beta}[\boldsymbol{x}_t, \boldsymbol{h}_{t-1}, \boldsymbol{m}_{t-1}] + \boldsymbol{b_\beta}) \\
\boldsymbol{v}_t^w &= \boldsymbol{f}(\boldsymbol{W_w}[\boldsymbol{x}_t, \boldsymbol{h}_{t-1}, \boldsymbol{m}_{t-1}] + \boldsymbol{b_w}) \\
\boldsymbol{v}_t^e &= \boldsymbol{f}(\boldsymbol{W_e}[\boldsymbol{x}_t, \boldsymbol{h}_{t-1}, \boldsymbol{m}_{t-1}] + \boldsymbol{b_e}) \\
\boldsymbol{m}_t &= \boldsymbol{m}_{t-1} + \boldsymbol{\alpha}_t \boldsymbol{v}_t^w - \boldsymbol{\beta}_t \boldsymbol{v}_t^e
\end{aligned}
\tag{5}
$$

$\boldsymbol{f}$ in the above relations stands for the (optional) nonlinearity of NRU, which, in cases where we opt to keep it, is ReLU. $\boldsymbol{m}_t$ is the memory cell at time step $t$. Notice that the original NRU has $k$ number of scalars $\boldsymbol{\alpha}$, $\boldsymbol{\beta}$, and memory vectors $\boldsymbol{v}^w$ and $\boldsymbol{v}^e$, known as the write (w) and erase (e) heads. In our experiments, we only consider one head, which results in Equation (5).

We use the same notation $\boldsymbol{W}$ for all weights to avoid notation complications. However, it is important to note that the weights for all vector values, i.e., $\boldsymbol{h}_t$, $\boldsymbol{v}_t^w$ and $\boldsymbol{v}_t^e$ are matrices, whereas the weights in the update equations of the scalar values $\boldsymbol{\alpha}_t$ and $\boldsymbol{\beta}_t$ are vectors.

Finally, regarding the memory mechanism of this architecture, the non-saturating activation function facilitates the flow of information compared with the saturating nonlinearity in LSTM and GRU. Similar to LSTM, NRU has a memory cell, though of arbitrary size, that provides an additional pathway for information flow.

### A.2 Transformers

Transformers are a type of neural network extensively used to analyze sequential data. Unlike RNNs, Transformers process data in parallel rather than sequentially. This parallel processing makes the analysis of long sequences much faster. Moreover, Transformers do not suffer from the vanishing/exploding gradient problem, allowing them to capture long-range dependencies more effectively. This efficacy is achieved through the self-attention mechanism, which enables the Transformer model to focus on the important parts of the sequence for each task. The attention mechanism computes attention scores for each step in the sequence; these attention scores are used to compute a weighted sum of the values, producing the output of the self-attention mechanism

$$\text{Attention}(\boldsymbol{Q}, \boldsymbol{K}, \boldsymbol{V}) = \text{softmax}\left(\frac{\boldsymbol{Q}\boldsymbol{K}^T}{\sqrt{d_k}}\right)\boldsymbol{V} \tag{6}$$

Following this, a feedforward network (FFN) is applied to the attention module's output in a position-wise manner. This feedforward module consists of a two-layer multilayer perceptron (MLP) with a ReLU nonlinearity between the layers

$$\text{FFN}(\boldsymbol{x}) = \boldsymbol{W}_2 \, \text{ReLU}(\boldsymbol{W}_1 \boldsymbol{x} + \boldsymbol{b}_1) + \boldsymbol{b}_2 \tag{7}$$

Despite the widespread use of Transformers across various tasks and data modalities, a significant concern remains the costly self-attention mechanism. The computational demand for attention scores scales quadratically with the sequence length, leaving room for the development of more efficient models, such as State-Space Models (SSMs), which still offer parallelization. There is also extensive literature on lower-complexity Transformers that mitigate this concern, but it is outside the scope of our work. (See (Fournier et al., 2023) for a review of such architectures.) A Transformer does not demonstrate the same sequential properties of recurrent networks and hence does not have well-defined states as recurrent networks do, thus the model does not need to memorize information in its state. Instead, at each step of the sequential data, the model can access any token coming before the current step independently of how far that token is.

### A.3 State Space Models

Recurrent neural networks still remain difficult to train and adapt, leading to a new paradigm of models called *state-space models*, borrowing from a fundamental concept from control theory. In control theory, state-space models map an input $\boldsymbol{x}(t) \in \mathbb{R}^d$ to an intermediate state $\boldsymbol{h}(t) \in \mathbb{R}^n$ that is further projected to an output $\boldsymbol{y}(t) \in \mathbb{R}^d$ using:

$$\begin{cases} \boldsymbol{h}'(t) &= \boldsymbol{A}\boldsymbol{h}(t) + \boldsymbol{B}\boldsymbol{x}(t) \\ \boldsymbol{y}(t) &= \boldsymbol{C}\boldsymbol{h}(t) + \boldsymbol{D}\boldsymbol{x}(t) \end{cases} \tag{8}$$

where $\boldsymbol{A}$, $\boldsymbol{B}$, $\boldsymbol{C}$ and $\boldsymbol{D}$ are all trainable parameters. Gu et al. (2021) use this paradigm to define a recurrent model to work on discrete signals, in which case the input can be regarded as discretized data sampled from a continuous signal with a step size $\Delta$, for which the corresponding SSM is defined by:

$$\boldsymbol{h}_t = \overline{\boldsymbol{A}}\boldsymbol{h}_{t-1} + \overline{\boldsymbol{B}}\boldsymbol{x}_t \qquad \qquad \boldsymbol{y}_t = \overline{\boldsymbol{C}}\boldsymbol{h}_t + \overline{\boldsymbol{D}}\boldsymbol{x}_t$$
$$\text{where:} \quad \overline{\boldsymbol{A}} = \left(I - \frac{\Delta}{2}\boldsymbol{A}\right)^{-1}\left(I + \frac{\Delta}{2}\boldsymbol{A}\right) \qquad \overline{\boldsymbol{B}} = \left(I - \frac{\Delta}{2}\boldsymbol{A}\right)^{-1}\Delta\boldsymbol{B} \tag{9}$$

and $\overline{\boldsymbol{C}} = \boldsymbol{C}$ ($\overline{\boldsymbol{D}}$ is a residual connection and set to $\boldsymbol{0}$ in most settings). Thus

$$\boldsymbol{y} = \sum_{j=0}^{L-1} \overline{\boldsymbol{C}}\overline{\boldsymbol{A}}^j\overline{\boldsymbol{B}}\boldsymbol{x}_{L-j} = \overline{\boldsymbol{K}} * \boldsymbol{x} \quad \text{where:} \quad \overline{\boldsymbol{K}} = (\overline{\boldsymbol{C}\boldsymbol{B}}, \overline{\boldsymbol{C}\boldsymbol{A}\boldsymbol{B}}, \dots, \overline{\boldsymbol{C}\boldsymbol{A}}^{L-1}\overline{\boldsymbol{B}}) \tag{10}$$

$\overline{\boldsymbol{K}}$ is the SSM kernel and $*$ denotes the convolution operation.

Under this paradigm, the model is considered to have a *recurrent* representation, which is efficient for inference and autoregressive generation of tokens, and a *convolutional* representation, which is used for efficient training over a sequence in parallel.

### A.3.1 Structured State-Space Sequence Model

Using the state-space paradigm for sequence models , Gu et al. (2021) introduce the *linear state-space layer* (LSSL), which implements a recurrent model using the discretized formulation of a state-space model, as the backbone of the first state-space based sequence model.

One of the key aspects of the state-space model formulation for a recurrent sequence model is the parameter $A$, which captures information from the previous state $h_{t-1}$ to construct the new state $h_t$. This forms the bottleneck in how information is carried forward across time in the sequence. Gu et al. (2022b) noticed that despite improvements over traditional recurrent models, LSSL models still had difficulties with long sequences. To overcome this, they introduce a specific initialization of $A$, within a framework called High-order Polynomial Projection Operators (HiPPO) (Gu et al., 2020), to enable the state-space model to preserve long-term dependency information [5]. Another important component of their architecture design is the parameterization of the transition matrix A, which falls into the category of structured matrices, decomposable as the sum of a normal matrix and a low-rank matrix, and thus allows for efficient matrix multiplication. Interestingly, this class of structured matrices includes the HiPPO-based initialization of $A$. This structural property is crucial for the efficient calculation of the kernel matrix $\overline{K}$, making the convolutional mode of the SSM calculations viable. As a result, $y$ can be computed in $O((L+N)\log(L+N))$, with $L$ and $N$ denoting the sequence length and hidden state size, respectively, using a Fast Fourier Transform (Cormen et al., 2009). Therefore, the entire output can be computed in tandem based on the input, given the matrices that parametrize the system. This led to the Structured State-Space Sequence Model (S4), which demonstrated significant state-of-the-art improvements in long-sequence modeling tasks.

### A.3.2 Diagonal State-Space Models

Because the convolutional perspective of the SSM requires repeated matrix multiplications (necessary for the computation of the kernel $\overline{K}$), Gu et al. (2022a) suggest to use a diagonal $A$. This enables the model to retain its general long-sequence modeling capabilities while enabling a much more efficient computation of $\overline{K}$, as exponentiation of a diagonal matrix can be done trivially by raising the diagonal elements to the appropriate power. Thus, the authors introduce the Diagonal S4 (S4D) variant as a more efficient but equally effective alternative to S4.

### A.3.3 Mamba

One particularity of the S4 family of models is that they are Linear Time Invariant (LTI), meaning that $A$, $B$, and $C$ are fixed for all time-steps, as is the time-step size $\Delta$. This means each token is treated equally in terms of how it contributes to the hidden state or output. This can have disadvantages, primarily because S4 and S4D are unable to perform content-aware reasoning. This implies that S4 and its variants underperform on certain tasks crucial to language modeling and generation, especially those that assess the ability to focus on or disregard specific inputs.

To address the issue of linear time invariance, Gu & Dao (2024) introduce a modification to S4: a selection mechanism that adaptively adjusts the extent to which tokens contribute to the hidden state and output. For an input sequence $x = (x_1, x_2, \ldots, x_t)$, the matrices $B$, $C$ and the step-size $\Delta$ are made dependent on the input at each time-step, thereby creating multiple $B_i$, $C_i$, $\Delta_i$ used to compute the corresponding outputs $y_i$ and states $h_i$. $A$ meanwhile is kept constant between steps, as the underlying rate at which the state changes for a fixed duration in time should be independent of the input. The variable step sizes $\Delta_i$ further control how much to ignore the input and instead focus on the previous context, with smaller values causing the system to focus on the context and larger step sizes causing it to focus on the input. In summary, Mamba improves on S4 by defining

$$h_t = \overline{A}(x_t)h_{t-1} + \overline{B}(x_t)x_t \qquad\qquad y_t = C(x_t)h_t + D(x_t)x_t$$

$$\text{with:} \quad \overline{A}(x_t) = \big(I - \tfrac{\Delta(x_t)}{2}A\big)^{-1}\big(I + \tfrac{\Delta(x_t)}{2}A\big) \qquad \overline{B}(x_t) = \big(I - \tfrac{\Delta(x_t)}{2}A\big)^{-1}\Delta(x_t)B(x_t) \tag{11}$$

---

[5]The notion behind the HiPPO parametrization is that it produces a hidden state that memorizes the history of a sequence, done mathematically by tracking the coefficients of a Legendre polynomial.

where each input-dependent matrix is computed with a single linear layer.

Gu & Dao (2024) showed that an input-dependent time-step $\Delta_i$ has a similar effect as the gating mechanism in GRU and is the modification that enables it to attain better performance in tasks that require information filtering, such as the denoising task.

### A.4 Hyper-parameters Search

We conducted a grid search over six different learning rates: $1e-2$, $5e-3$, $1e-3$, $5e-4$, $1e-4$, $5e-5$, $1e-5$ and we chose the best performing model based on its extrapolation performance, rather than its best training behavior, such as fastest convergence. The learning plots show the result for that specific learning rate for a given model.

## B Experiment Details and Hyper-Parameter for State Tracking Tasks

On the state tracking tasks, we train each model for up to 500 epochs, with early stopping if performance reaches 90% accuracy, or fails to reach 1% accuracy after half the allocated training time. A model is deemed to learn the task if it achieves the threshold performance before the training budget elapses.

When training on sequences of length $k$, we first create a dataset of elements from the selected group of length $k$. We then randomly select a maximum of $10^6$ elements from the dataset, which is further split into a training and testing split. Following Merrill et al. (2024), we always include all examples for $k = 2$ in the training set.

When evaluating for extrapolation to longer lengths, the number of parameters per layer is kept constant between models. But, we use the minimal number of layers necessary to achieve the threshold accuracy on the training length (empirically chosen as 90% in our experimentas). This number of layers can vary between different models for each task. For the NRU, we use a memory dimension of 256, with ReLU activation. For the IDS4, we use a state dimension of 128. For the Transformer, we use 8 heads. We use a batch size of 256 and AdamW optimizer with a learning rate of $10^{-4}$, $(\beta_1, \beta_2) = (0.9, 0.999)$ and a Weight Decay of 0.01.

| Model | RNN | LSTM | GRU | NRU | Transformer | S4D | Mamba | IDS4 |
|---|---|---|---|---|---|---|---|---|
| $d_{\mathrm{emb}}$ | 256 | 256 | 256 | 256 | 512 | 1174 | 512 | 256 |
| $d_{\mathrm{hidden}}$ | 768 | 768 | 768 | 1024 | - | - | - | - |

Table 6: Model Hyper-parameters for State Tracking Tasks

## C Additional Results for State Tracking Tasks

Figure 4 shows accuracy on a held-out validation set during training. Clearly, the direct learning trends of the different models change both between tasks as well as length. For example, for sequences of length $k = 128$, the RNN manages to learn the task significantly faster than other models on the groups $\mathbb{A}_5$ and $\mathbb{Z}_{60}$, yet eventually the other models remain capable of learning on longer sequences, while the RNN cannot ($k = 256$ for $\mathbb{A}_5$ and $k = 512$ for $\mathbb{Z}_{60}$, respectively)[6]. This highlights the possibility that the inductive biases of such models, despite modeling the correct circuit complexity for these classes of problems, remain insufficient for such state tracking tasks.

---

[6]We conduct a grid search on a select set of hyper-parameters. Increasing model size (which we control for) may potentially change convergence rates of each model.

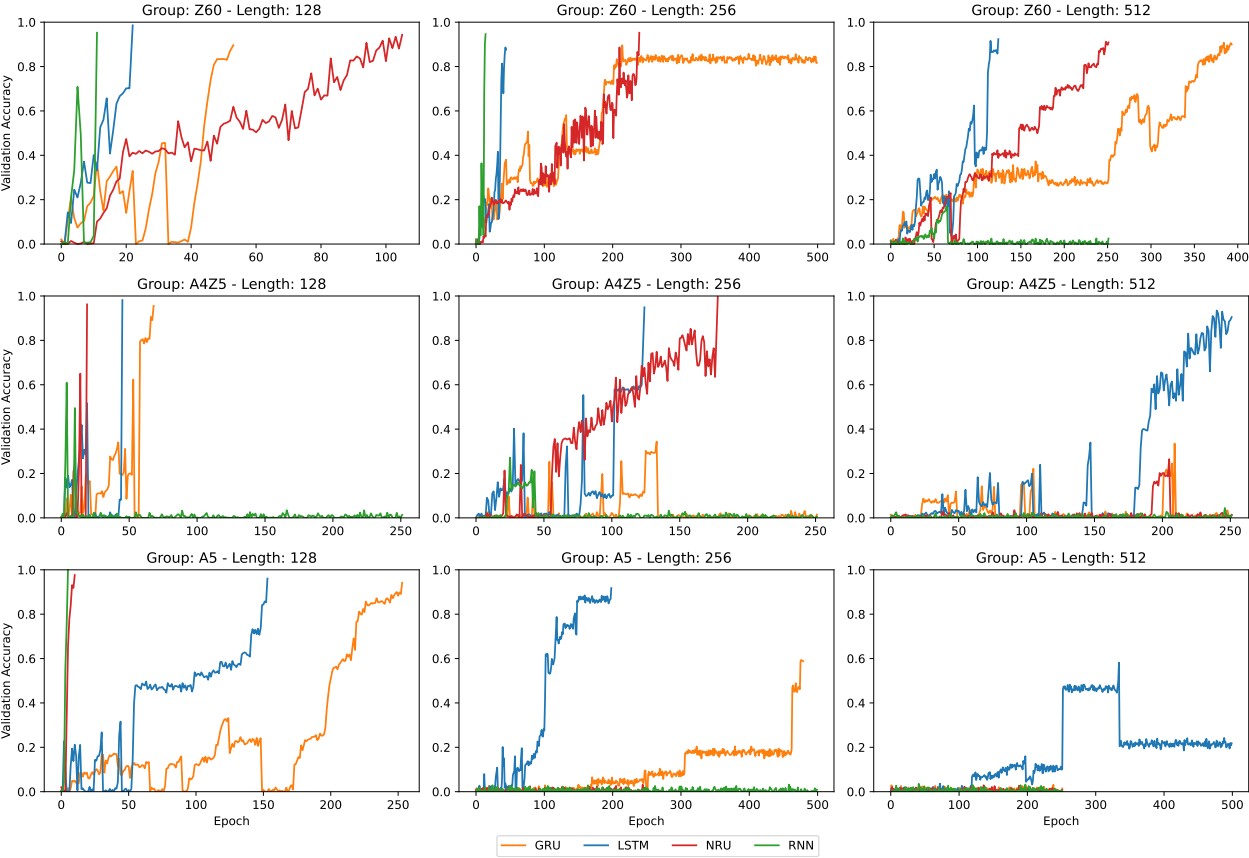

Figure 4: Ability of single-layer recurrent models to learn longer state tracking tasks. We observe that as the sequence gets increasingly long, even models that theoretically should be able to learn the task fail to do so, especially as the state tracking task becomes more complex (in terms of circuit complexity).

| Model | Hidden size for 1-layer model | Hidden size for 2-layer model |
|---|---|---|
| **RNN** | 216 | 200 |
| **LSTM** | 112 | 96 |
| **NRU** | 128 | 128 |
| **GRU** | 128 | 112 |
| **TRANSFORMER** | 64 | 64 |
| **Mamba** | 256 | 256 |
| **S4D** | 2400 | 3000 |

Table 7: Hidden size for different models with comparable sizes. Notice that for Mamba, 256 is the largest hidden size that could be set due to source code restrictions. Hence, Mamba is smaller than the rest of the models.

## D    Hyper-parameters for Copying and Denoising Tasks

- **Fixed hyperparameters** For training hyperparameters, we use Adam optimizer with learning rate $1e-3$. We use gradient clipping with value 1. The maximum number of steps is set to 150K, but training stops as the model reaches 99% accuracy. As for model hyperparameters, the input and output dimensions are equal to 10, which is the number of tokens in our vocabulary, with 8 values for the random digits in the sequence, one value for the blank token, and one for the indicator token. We use one more token for padding as we train on varying-length sequences. In cases where we

use embedding, the embedding size is also set to 10. The batch size for all models except for the Transformer is 2048. For the Transformer case, we use a smaller batch size of 1024. The maximum length of the random sequence is fixed to 10 and the maximum length of the noise sequence is 50. We use a mixture of sequences with various lengths of the random part of the sequences for experiments that will later be used to examine extrapolation to longer random sequence length and mixed noise interval length for experiments that will later be used to examine extrapolation to longer noise sequence length. In the first case, we consider a mixture of random lengths $2, 4, 6, 8, 10$, and in the second case, we have a mixture of noise lengths $10, 20, 30, 40, 50$.

- **Tuned hyperparameters** We run experiments for both 1- and 2-layer architectures. For the experiments with the same hidden size dimension, we use hidden size 128. For experiments where we want almost the same number of parameters for all architectures with the same number of layers, the hidden size values for different architectures are provided in Table 7.

# E   Additional Results on Copying Tasks

This section contains all the training dynamics plots for copying and denoising tasks when all models have similar number of parameters.

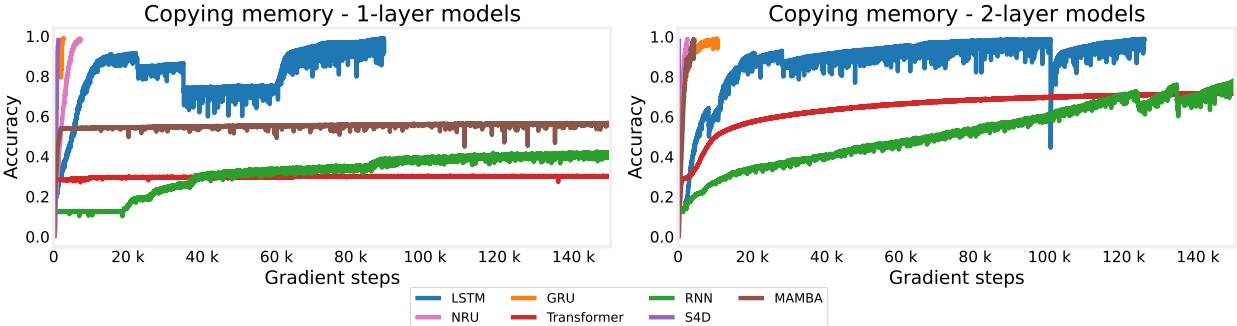

Figure 5: (Learning dynamics of (left) 1- and (right) 2-layer models with **similar number of parameters** for **copying memory** task trained on sequences with a **mixture of blank interval lengths**.

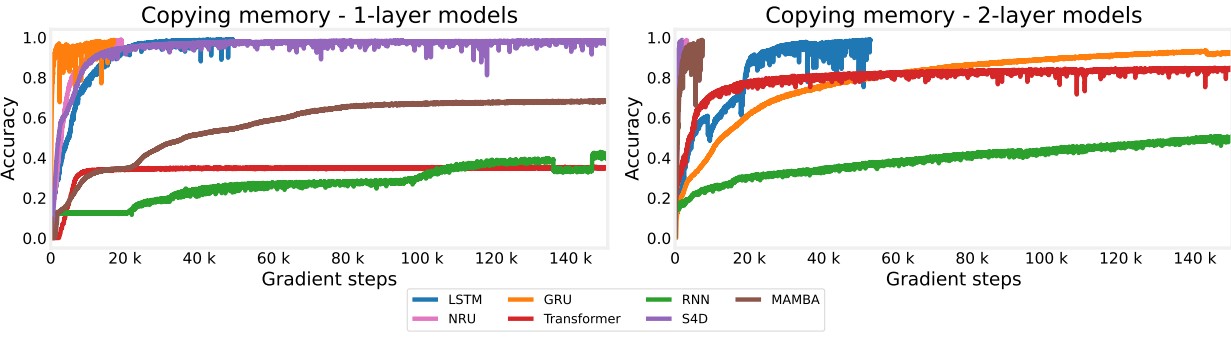

Figure 6: Learning dynamics for **copying memory** task with **similar number of parameters** trained on sequences with a **mixture of pattern lengths** up to 10. We observe that all gated RNNs, as well as S4D, are able to solve the task with only one layer, while Mamba and Transformer need a second layer.

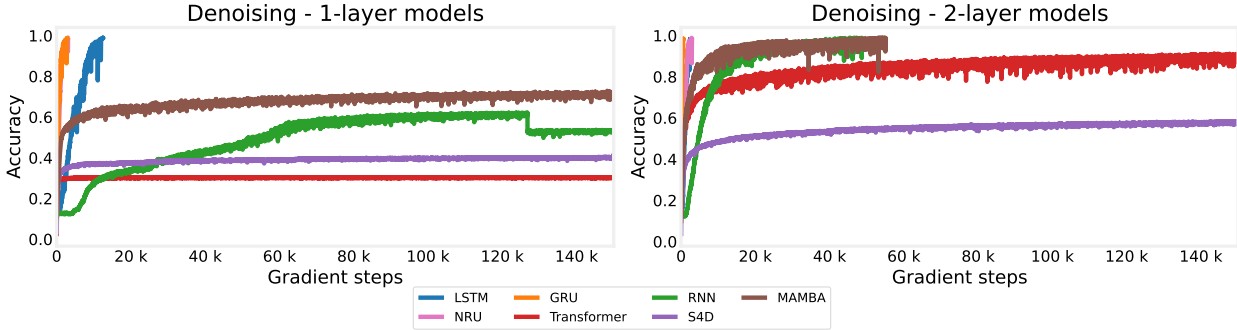

Figure 7: Learning dynamics for **denoising** task with **similar number of parameters** across models trained on sequences with a **mixture of blank interval lengths**. With a larger size, S4D still has a very slow performance even with 2 layers.

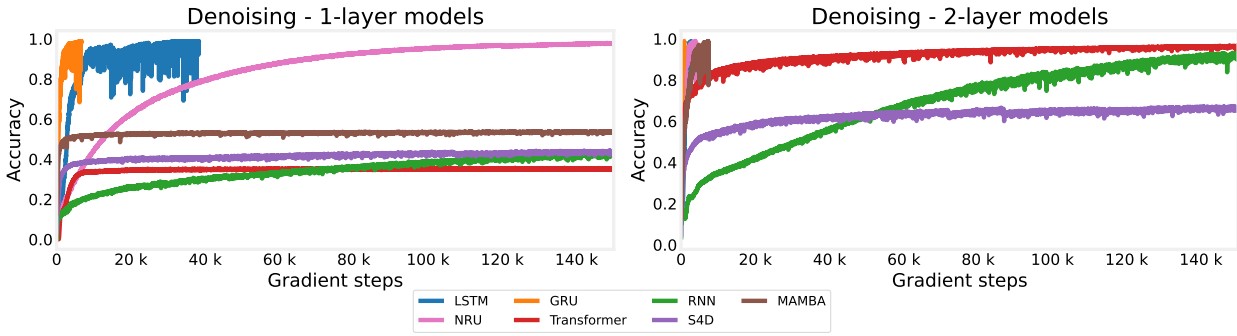

Figure 8: Learning dynamics for **denoising task** for models with **similar number of parameters** trained on sequences with a **mixture of pattern lengths**. While 2-layer Mamba solves the task, S4D shows a very slow performance. Again, RNN models beat all other architectures.

# F  Additional Results on Copying Tasks

This section is devoted to the results for Copying and Denoising tasks where different models have the same hidden size. The overall observations are similar to the experiments where models have similar number of parameters.

## F.1  Copying Memory

### F.1.1  Mixed Blank Interval

Figure 9 and Figure 10 show the learning dynamics and extrapolation performance of models with the same hidden size trained on samples of mixed blank interval length on the copying task.

### F.1.2  Mixed Pattern interval

Figure 11 and Figure 12 show the learning dynamics as well as extrapolation performance of models with the same hidden size trained on samples of mixed pattern lengths on the copying task.

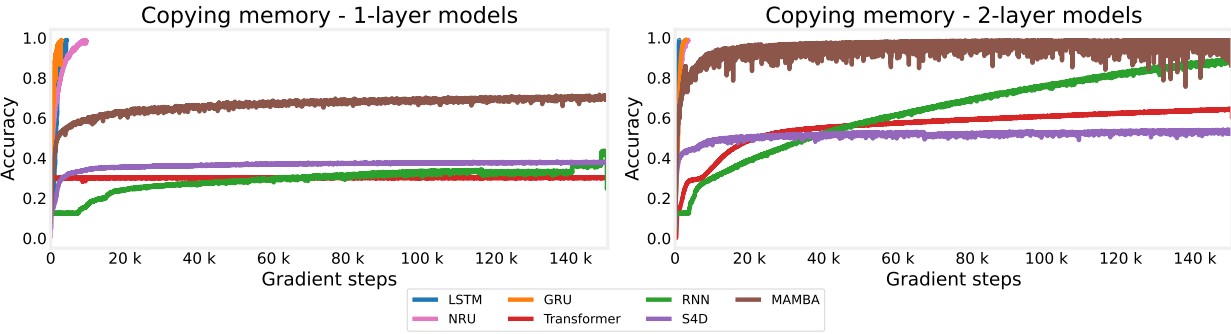

Figure 9: Learning dynamics for **copying memory** task with the **same hidden size** 128 trained on sequences with a **mixture of blank interval lengths**. 1-layer S4D and gated RNNs solve the task, while 2-layer Mamba and Transformer show much slower performance.

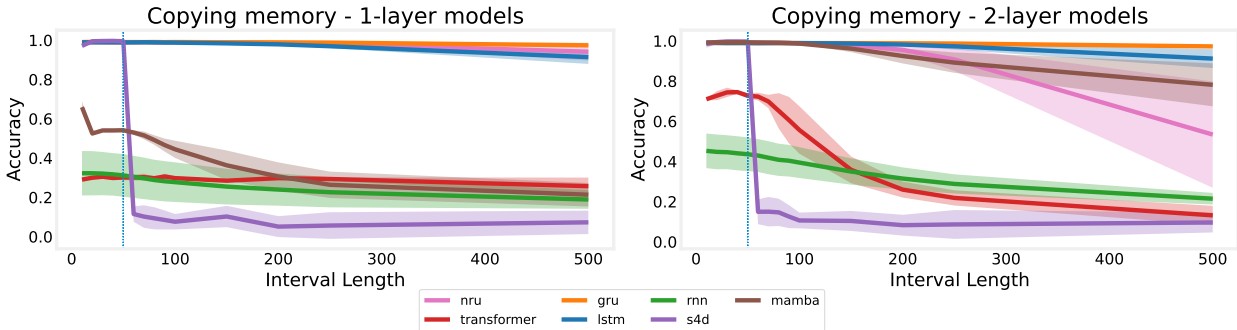

Figure 10: Extrapolation of (left) 1- and (right) 2-layer models with the **same hidden size** to longer **blank interval length** for **copying task**. The dashed vertical blue line is the training range.

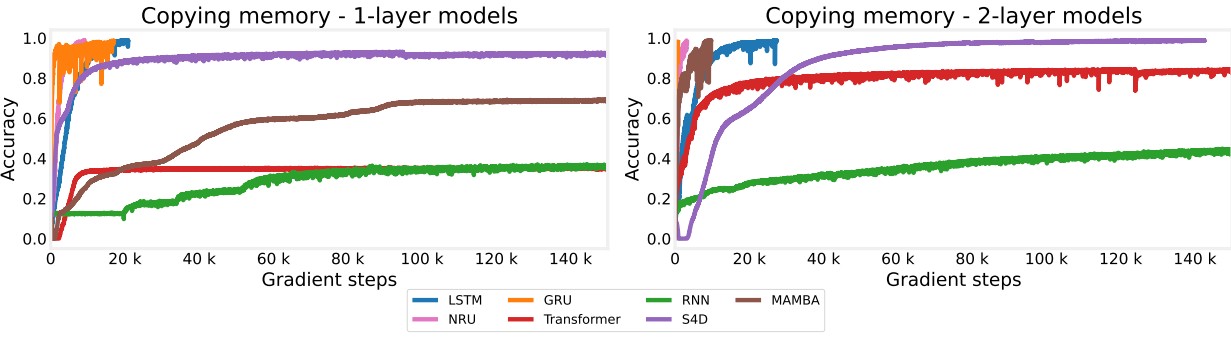

Figure 11: Learning dynamics for **copying memory** task for models with **same hidden sizes** trained on sequences with a **mixture of pattern lengths**. Here, we have similar observations to the case with similar model sizes in terms of models ability to achieve perfect performance. Compared with the case with same number of parameters, S4D has slower performance, which can be explained by its much smaller size when restricted to have the same hidden size as other models.

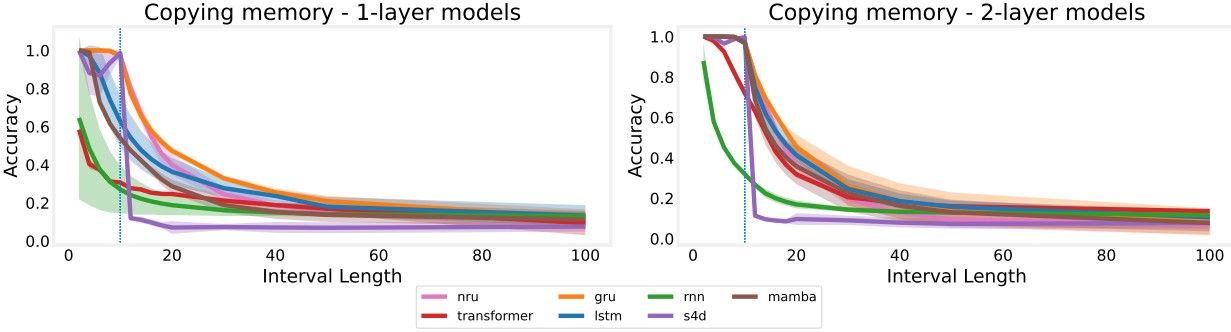

Figure 12: extrapolation of (left) 1- and (right) 2-layer models with the **same hidden size** to longer **pattern length** for **copying task**. Similar to the case with comparable model sizes, all models fail to extrapolate for pattern lengths twice the maximum training length and beyond.

## F.2 Denoising

### F.2.1 Mixed Interval Lengths

Figure 13 and Figure 14 show the learning dynamics and extrapolation performance of models with the same hidden size trained on samples of mixed blank interval length on the denoising task.

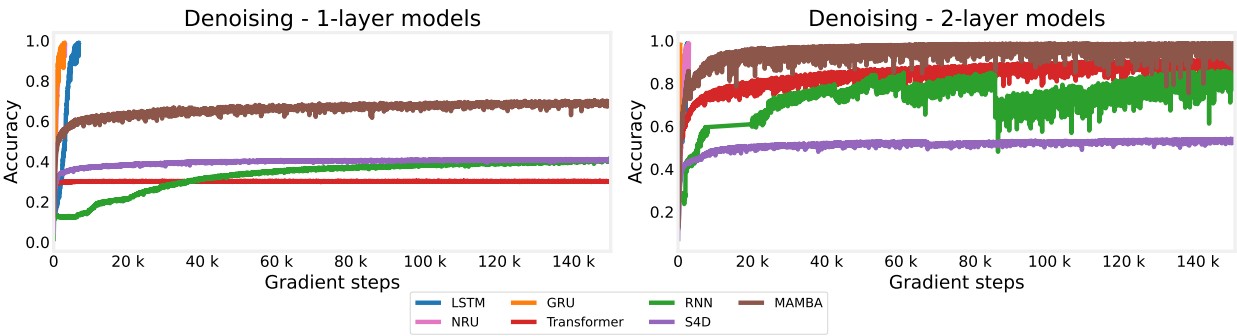

Figure 13: Learning dynamics for **denoising task** with the **same hidden size** trained on sequences with a **mixture of blank interval lengths**.

### F.2.2 Mixed Pattern Lengths

Figure 15 and Figure 16 show the learning dynamics as well as extrapolation performance of models with the same hidden size trained on samples of mixed pattern lengths on the denoising task.

## G  Data Generation for Counting Task

The dataset for training the models is a synthetic dataset generated using Python. This function generates sequences of scalars that alternate between *off* (represented by zeros) and *on* (represented by ones) states as described above. The *off* period lasts for a random number of time steps, chosen between a specified range (ITI), while the *on* period has a fixed length (ISI) which is predetermined. The total length of each sequence is capped at $n$ time steps, and the function generates $k$ sequences in this manner. For each sequence, the function prepares input and target values, where the target is simply a shifted version of the input (a time step behind) to do the next input prediction.

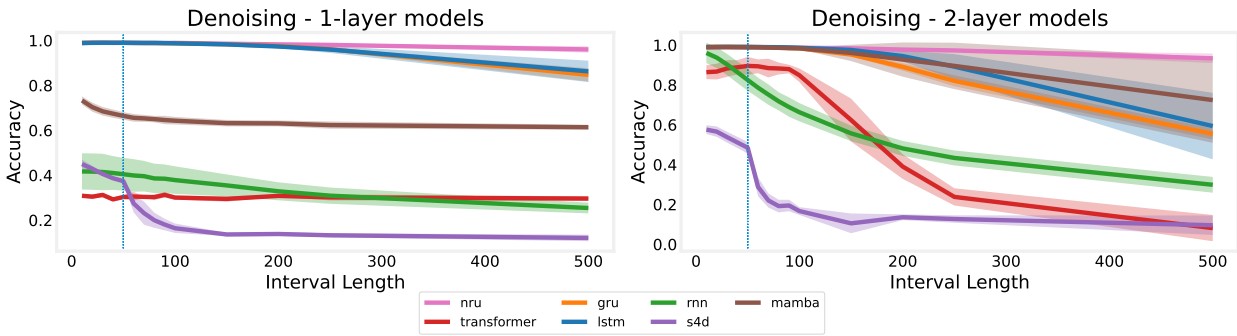

Figure 14: Extrapolation of (left) 1- and (right) 2-layer models with the **same hidden size** to longer **blank interval** interval length for **denoising task**. Similar to the copying task, we observe that while S4D cannot extrapolate to longer sequences, 2-layer Mamba resides more in the category of gated RNNs.

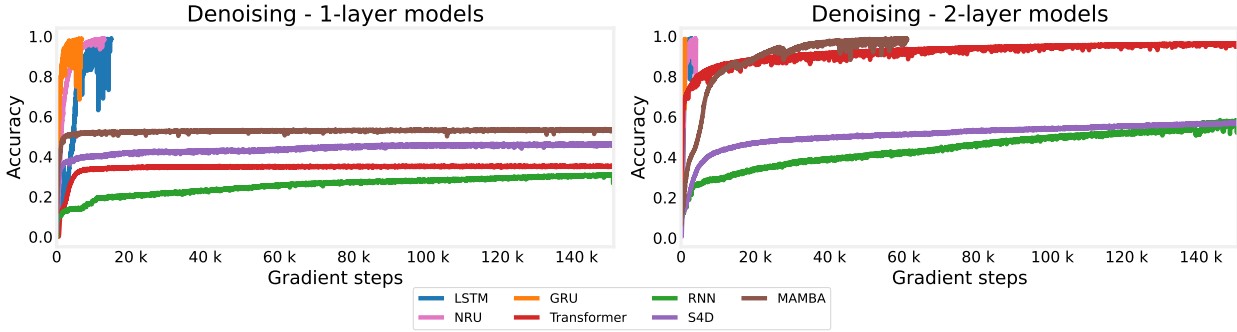

Figure 15: Learning dynamics for **denoising task** for models with **same hidden sizes** trained on sequences with a **mixture of pattern lengths**.

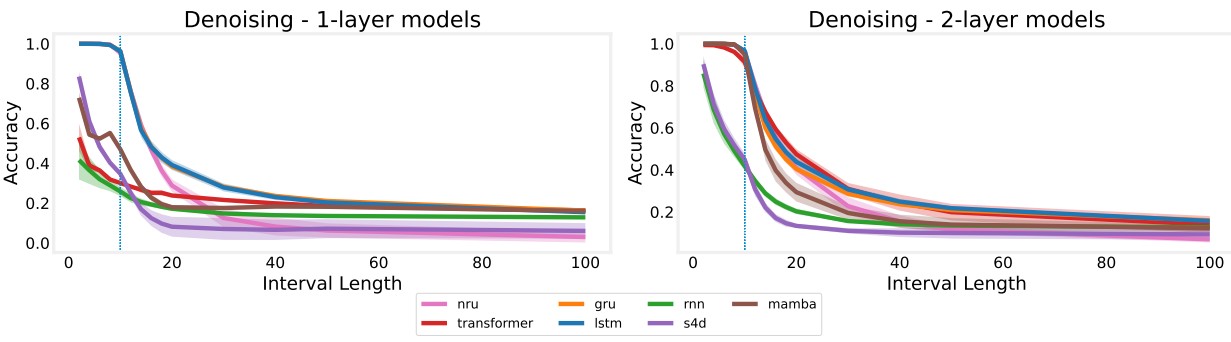

Figure 16: extrapolation of (left) one and (right) two-layer models with the **same hidden size** to longer **pattern** length for **denoising task**.

## H  Model Architectures for Counting Tasks

For RNN, LSTM, and GRU, we use the default PyTorch implementation, augmented by the optional skip connections and normalization layers between layers. For the Transformer, we use a PyTorch implementation of a Transformer decoder where each layer is set to only have access to the tokens coming before the token to be predicted. This causal architecture ensures that the Transformer model is learning the task while having

| Hidden Size 8 | |
|---|---|
| Model | Total Parameters |
| GRU | 275 |
| LSTM | 363 |
| Mamba | 3641 |
| NRU | 3205 |
| RNN | 99 |
| S4D | 1225 |
| Transformer | 2529 |

| Hidden Size 64 | |
|---|---|
| Model | Total Parameters |
| GRU | 12931 |
| LSTM | 17219 |
| Mamba | 51393 |
| NRU | 17093 |
| RNN | 4355 |
| S4D | 16961 |
| Transformer | 33665 |

Table 8: Number of Model Parameters.

access to the same information as the other models in our experiment, which are all recurrence-based. Also, we consider two different positional encoding methods: NoPE (or no positional encoding) and the traditional sine-cosine relative positional embedding.

As for the state-space models, S4D and Mamba, we use the same architectures as in their original paper, augmented by skip connection and optional pre- or post-layer normalization.

## I   A Solution for Offset Prediction Task with S4D

We consider the task of next-token prediction for 1s and random bursts of 0s of fixed length $n$. We show that one layer of S4D with complex scalar state is capable of correctly predicting all tokens except for the first token of a burst. At a high-level, we will be simulating a finite-state automaton that remains in a *sleep* state when observing 1s, and *counts* to $n$ when observing 0s, ending up back at the sleep state. There are thus $n$ states in total. Note that the automata will not be able to handle interruptions to the counting procedure, so it will go into a fail state if the number of 0s is not a multiple of $n$. The model will predict 1 at the sleep state and 0 otherwise. Letting $h_0 = 1, A = \exp(i2\pi/n), B = 1 - \exp(i2\pi/n)$ constructs the desired system. The sleep state is $h = 1$. The output function is $\phi(h_t) = \mathbf{1}_{\{h_t=1\}}$.

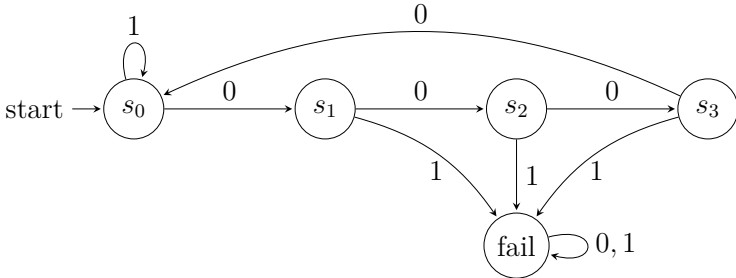

Figure 17: Finite-state automaton for offset prediction task when $n = 4$.

