# OpenReview forum: "On Memory and Generalization in the Era of Linear Recurrence"
_TMLR — Rejected by TMLR_

### Review · Reviewer_nk7E · 2025-03-03

**Summary Of Contributions:**

The paper examines memory properties of sequence models, with particular emphasis on recently developed linear recurrent neural networks. It introduces a taxonomy of different memory types (state-less, stateful, stable, and counting) that are intended to be better suited to neural networks than definitions based on formal language theory. The paper then presents an empirical study across various tasks that examine different aspects of memory capabilities.

**Audience:**

Yes

**Broader Impact Concerns:**

This is a fundamental study, I don't have any concerns regarding the ethical implications of the work.

**Claims And Evidence:**

No

**Requested Changes:**

Perhaps most importantly, the learning rates need to be tuned for each method, and reporting results need to be reported on several (fresh) seeds given the high variability of experiments involving RNNs.

As a reader, I would expect a paper on this topic to address several key questions that aren't fully explored:
1. How do specific architectural features affect different types of memory capabilities? Similarly, how do these features influence length generalization across different memory types?
2. How does the number of hidden units required to solve a task scale with the memory requirements? For example, a linear RNN with infinite precision could theoretically solve a simple counting task with a single hidden neuron, whereas a finite state machine would need a logarithmic number of states. An analysis of which kind of solutions gradient descent finds would add value to the paper.
3. Specific to linear RNNs: what impact does the matrix-valued hidden state of Mamba have? Does it substantially increase the model's memory capacity? Targeted ablations would help clarify this point.
While we can partially answer these questions from the results presented in the paper, a more thorough analysis and some additional experiments would strengthen the work considerably. I would additionally suggest starting the results section with a half-page summary of key findings, with the remainder dedicated to providing supporting evidence. This would help readers quickly grasp the main conclusions, which is currently challenging.

In its current form, the paper is longer than necessary. Important findings are interspersed with technical details, making it difficult to identify the key contributions. Specifically:
- The introduction reads partly as an extensive literature review rather than focusing on motivating the study.
- Standard experimental details occupy considerable space.
- Much of the analysis provides extensive but relatively superficial descriptions of the results.
I would recommend condensing the paper to approximately 10-12 pages, moving detailed experimental methods and literature review to the appendix.

Regarding the taxonomy, I would appreciate clarification on why counting and stable memories are treated differently from stateful memories. The distinction is clear in formal language theory, but appears less evident in the neural network context (at least for finite sequences), cf. the example mentioned in question 2. above. This may reflect my misunderstanding rather than an issue with the paper.

**Strengths And Weaknesses:**

**Strengths:**
- The paper is well-written and clearly structured.
- The comparison to existing work is thorough and demonstrates good awareness of the field.
- The set of experiments picked nicely aligns with the taxonomy.

**Weaknesses:**
- While the memory taxonomy is presented as being more applicable to neural networks, it still appears very close to automata theory. This creates distinctions between memory types that may not align with (my) intuition (cf. next section for more detail).
- The learning rates do not appear to be tuned for each architecture and a single seed is used. This considerably limits the robustness of the conclusions of the experiments.
- The paper would benefit from being more concise - there is repetition across sections that could be condensed.
- The novelty is somewhat limited, as these tasks have been extensively studied in prior work. The analysis of the results is rather descriptive, and does not currently provide an in-depth understanding of the abilities of different models (cf. next section for suggestions to improve it).

**Minor:**
- Terminology inconsistency: the selective copy task is sometimes referred to as "denoising". Using consistent terminology would improve clarity.

---

> ### Author Response · Authors · 2025-07-15
> **Author Response (1/n)**
>
> We sincerely appreciate the reviewers’ detailed feedback and valuable suggestions. We are pleased that they found our comprehensive comparison of memory and generalization in RNNs, SMMs, and Transformers to be a novel contribution (Jfrh, pQWm, nk7E) and our paper well-written (pQWm, nk7E). We have carefully considered the reviewers’ comments and concerns and have addressed them below.
>
> -----
>
> > **Requested Changes**
>
> **Comment.** Perhaps most importantly, the learning rates need to be tuned for each method, and reporting results need to be reported on several (fresh) seeds given the high variability of experiments involving RNNs.
>
> **Answer.** We have thoroughly conducted a grid search over a range of learning rates: {1e-5, 5e-5, 1e-4, 5e-4, 1e-3, 5e-3, 1e-2}, ensuring a comprehensive evaluation of potential impacts on our final observations. Details were added in Appendices A.4 onwards. The grid search did not significantly alter our ability to solve tasks or extrapolate, but it did influence the convergence speed of the models.
>
> We believe there has been a misunderstanding about the number of seeds. We used 5 seeds and averaged the results over them (or for some cases considered the good performance of at least 3 seeds out of 5 as success in a task) in all experiments, which has now been clarified in the updated paper.
>
> **Comment.** 1-How do specific architectural features affect different types of memory capabilities? Similarly, how do these features influence length generalization across different memory types?
>
> **Answer.** We agree that the memory capabilities of a model depend on its architectural features. For instance, the attention mechanism in transformers effectively results in an infinite-context memory, allowing the model to attend to any past token without a fixed-size bottleneck. Consequently, transformers are well-suited for and should excel at tasks such as copying or denoising, as far as the concern is memorization capacity. On the other hand, in tasks like state-tracking that involve a limited number of possible states (e.g., Z60 resembles moving around a circle with 60 positions), RNNs should theoretically be able to solve the problem, provided that their hidden state is sufficiently large to map these potential states to distinct numerical values and that the true function solely depends on the current input and state and not past inputs.
>
> Generalization and extrapolation also depend on the task and the model’s architectural features. For state-tracking, we hypothesize that solving the task itself is sufficient for extrapolation. However, this is not the case for counting, as the model may need to learn to accumulate and reset in ways that were not seen during training. For tasks like copying or denoising, transformers may extrapolate better due to the absence of a recurrent state that restricts information flow between time steps.
>
> To this end, we have also extended our analysis at the end of each experimental Sections 4.1, 4.2 and 4.3.

---

> ### Author Response · Authors · 2025-07-15
> **Author Response (2/n)**
>
> **Comment.** 2-How is the number of hidden units required to solve a task scale with the memory requirements? For example, a linear RNN with infinite precision could theoretically solve a simple counting task with a single hidden neuron, whereas a finite state machine would need a logarithmic number of states. An analysis of which kind of solutions gradient descent finds would add value to the paper.
>
> **Answer.** This is a very interesting question, unfortunately we do not have extensive experiments on the number of hidden units and we believe that it is a hard question to answer theoretically. Nevertheless,  we would like to emphasize that in the paper we have the results for  two different hidden sizes for the counting task, i.e.,  8 and 16. As expected from the simplicity of this task, 8 was generally enough and 16 sometimes even hurt the performance. Note that while we did not analyse which kind of solution GD has found, we have added an analytical solution in Appendix I for linear RNN, which as pointed out by the reviewer reveals the sufficiency of one single hidden unit for this task, yet in practice we see that larger hidden size is required for the model to find any solution. Also, our extrapolation results are meant as a complementary manner of verifying the correct solution learned; the ability to extrapolate reveals (at least partially) to what extent the solution found by the model matches the general solution that would be necessary to solve the underlying task.
>
> Another point that makes it less straightforward to interpret the relationship between memory requirements and hidden state size for a general task is that the memory requirement is not the only or the main factor here, the computational difficulty of the task also plays an important role. For a simple task like counting, we can do such an analysis, because there is not much that is required to do by the hidden state other than keeping track of its counts. For state tracking on the other hand, the model should also use a large enough hidden state to learn an expressive enough transition function between the states. Note that there exists a direct relationship between state hidden size (d),  and transition matrix size (dxd). Hence, the larger the hidden size, the more parameters for the model.
> So, while state tracking is not a memory-intensive task, learning this transition function, i.e., group operations, can require large hidden sizes. But this large size should not be seen as an indication of high memory requirements.
> This being said, in our experiments, we opted for more or less the same number of hidden units used for the same or similar tasks in the literature.
> For state-tracking, we used a size comparable to [Merrill et al., 2024]. For copying, we used similar hidden sizes as in [Chandar et al., 2019]. Finally, for counting, the task itself is relatively easy, as 8 hidden units was sufficient.
>
> One final point about hidden size in SSMs like Mamba vs. in RNNs: in Mamba, the construction is not exactly the same as with standard recurrent neural networks, as each dimension in fact uses a different hidden state (what is called a SISO-type SSM, and hence the resulting matrix-valued hidden state). As such, it is not an apples-to-apples comparison with usual RNNs in terms of how to conduct a hyperparameter search here, but we can generalize this such that the dimension times hidden state size is more or less similar to the RNN’s hidden size.
>
> As such, we did conduct a small ablation with per-dimension hidden state sizes of 8 and 16 (due to restrictions with the construction of the model) in our state tracking experiment, where the observations are the same; Mamba models fail to extrapolate to longer lengths while their learning of longer lengths remains limited (for single-layer models).
>
> All being said, we acknowledge that this question is very important to address. We believe that the reason we cannot give a more direct answer at this point is the difficulty of disentangling the contributions of both memory requirements and state transition complexity to the hidden size, that is beyond the scope of the current work. The focus here is more on: given a large enough hidden state so that the task can be solved, can the trained model generalize? However, this question for sure guides the next steps of our research on this topic.
>
> -----
>
> **References**
>
> Merrill et al., The Illusion of State in State-Space Models, International Conference on Machine Learning, 2024
>
> Chandar et al., Towards non-saturating recurrent units for modelling long-term dependencies,
> Proceedings of the AAAI Conference on Artificial Intelligence, 2019

---

> ### Author Response · Authors · 2025-07-15
> **Author Response (3/n)**
>
> **Comment.**  3-Specific to linear RNNs: what impact does the matrix-valued hidden state of Mamba have? Does it substantially increase the model's memory capacity? Targeted ablations would help clarify this point. While we can partially answer these questions from the results presented in the paper, a more thorough analysis and some additional experiments would strengthen the work considerably.
>
> **Answer.**
> In terms of empirical observations, as pointed out in item 2 in the summary of our results for copying and denoising tasks at the end of section 4.2, the matrix-valued hidden state of Mamba does not seem to improve its extrapolation on our most memory-intensive task, copying, compared to ordinary recurrent networks with vector-valued hidden state. This implies that this mechanism of memory augmentation is still not helpful with promoting the model on the Chomsky hierarchy. Furthermore,  theoretically, we believe that the matrix-valued state of Mamba which is a result of it having separate per-dimension hidden states, can be equivalently reshaped into a larger vector hidden state; so it doesn't explicitly change the memory structure in comparison to other RNNs (though there may be direct interactions that change within this structure).
>
> **Comment.** I would additionally suggest starting the results section with a half-page summary of key findings, with the remainder dedicated to providing supporting evidence. This would help readers quickly grasp the main conclusions, which is currently challenging.
>
> **Answer.** We appreciate the suggestion. We added a short summary of key findings at the beginning of section 4 and refined the last paragraph of each experiment section to summarize our main findings more concisely.

---

> ### Author Response · Authors · 2025-07-15
> **Author Response (4/n)**
>
> **Comment.** Regarding the taxonomy, I would appreciate clarification on why counting and stable memories are treated differently from stateful memories. The distinction is clear in formal language theory, but appears less evident in the neural network context (at least for finite sequences), cf. the example mentioned in question 2. above. This may reflect my misunderstanding rather than an issue with the paper.
>
> **Answer** Thank you for bringing up this interesting point. First of all, we completely agree that the distinction between the three cases is less evident when dealing with finite sequences. The same also applies to automata, as bounded sequences imply a limited number of states, resulting in a finite state automaton (FSA), which is known to be capable of modeling the task.
>
> We would like to emphasize, however, that this is actually the main focus of our length extrapolation experiments: to explore the unbounded state case, for which a finite-state automata would not suffice. The aim of our experiments, particularly the extrapolation component, is to show that there is strong evidence -- at least practical, if not theoretical -- on the benefits and limitations of each architecture.
>
> While we agree that, in principle, a vanilla RNN can allocate portions of its hidden state to function as additional memory, numerous empirical and theoretical studies have demonstrated the practical benefits of incorporating explicit memory mechanisms, especially for memory-intensive tasks such as copying.For instance, the seminal work by [Deletang et al., 2023] shows that scaling up architectures without extra memory—such as by adding more layers—does not enhance their ability to extrapolate to longer sequence lengths.  Another compelling study by [Wang et al., 2019] reveals that even models equipped with memory modules require more nuanced architectural modifications to achieve robust extrapolation. Specifically, they introduce state regularization, a subtle adjustment to the RNN architecture that guides the network toward learning a finite set of interpretable states.
>
> Assuming that the optimal solution for extrapolation corresponds to an automaton-like model with such states, it becomes crucial to equip the network with mechanisms that enable accurate state learning. Training dynamics appear to play a significant role in this process. Thus, at least in practice, neural networks with varying degrees of explicit memory capacity exhibit meaningful differences in performance. That said, we acknowledge that, for the specific counting task addressed in our paper, the distinction may be less pronounced. Due to the particular design of this experiment—in which the ISI component of the signal maintains a fixed length—a finite-state automaton (FSA) can solve the task (see Appendix I). Therefore, this example may not be the most representative case to support our broader taxonomy. We are grateful to the reviewer for highlighting this subtle but important point.
>
> In light of this question, we did add some clarifications to the taxonomy of counting section, more precise relationships with architectures at the end of each experimental section, and the finite state automaton (our new appendix I) that a simple linear RNN, like S4D can find for our counting task..
>
> So, in short, yes, we agree that the continuous nature of neural network state, especially in infinite precision, compared to the  automata with discrete states, makes the distinction between different memory requirements less evident; but on the other hand, contrary to the automata, in neural networks there is this additional step of learning those states and transitions between them which makes things less straightforward in another way.
>
> -----
>
> **References**
>
> Deletang et al., Neural Networks and the Chomsky Hierarchy, International Conference on Learning Representations, 2023.
>
> Wang et al., State-Regularized Recurrent Neural Networks, International Conference on Machine Learning, 2019

---

> ### Author Response · Authors · 2025-07-15
> **Author Response (5/n)**
>
> > **Other comments**
>
> **Comment.**  While the memory taxonomy is presented as being more applicable to neural networks, it still appears very close to automata theory.
>
> **Answer.** We would like to clarify that our contribution is not in understanding RNNs (and other sequence models) from the perspective of formal languages, but rather attempting to better categorize models based on how they structure data in the form of a “memory” and better relate this to the types of tasks that they can solve. We did add some more precise links with architectures in the taxonomy of counting section, at the end of each experimental section, as well as in Appendix I.
>
> **Comment.** The learning rates do not appear to be tuned for each architecture and a single seed is used. This considerably limits the robustness of the conclusions of the experiments.
>
> **Answer.**  As mentioned above, we have provided an additional learning rate search over the following rates {1e-5, 5e-5, 1e-4, 5e-4, 1e-3, 5e-3, 1e-2} to use the better learning rate over the experiments.Furthermore, there may be some confusion over the number of seeds used; for all experiments we used 5 seeds and averaged over them. We clarified that in the paper.
>
> **Comment.** The paper would benefit from being more concise - there is repetition across sections that could be condensed.
>
> **Answer.**  We agree that some parts of the paper can be better condensed, which we provide in an updated version of our submission. In particular:
>
> - We removed redundancies between the introduction and the related works section.
> - We moved some details of the descriptions into the Appendix.
> - We narrowed down some of the verbose descriptions and results to better focus on the provided tables and figures.
> - We moved some figures that demonstrate redundancies in the observations to the Appendix.
>
> **Comment.** The novelty is somewhat limited, as these tasks have been extensively studied in prior work. The analysis of the results is rather descriptive, and does not currently provide an in-depth understanding of the abilities of different models (cf. next section for suggestions to improve it).
>
> **Answer.** We included corrections for all the above suggestions and we extended our analysis at the end of each experimental Sections 4.1, 4.2 and 4.3 and in Appendix I to better relate each architecture’s results with its structure and internal limitations. We believe this new version of the paper delivers more insights. As mentioned by reviewer Jfrh, there was not such general studies in the literature that goes through a comparison of linear and full RNNs on the exact same set of tasks, the most overlapping one is [Grazzi et al., 2024] which was a concurrent work to ours and studies these models on state tracking tasks.  While our results with S4D and Mamba on state tracking are not new in light of this extensive study, the results are consistent to their theory and empirics; moreover, we have some additional state tracking experiment with another model from [Merrill et al., 2024] called IDS4, that to the best of our knowledge, was not included in any earlier studies of generalization. As pointed out in item 2 of our results at the end of section 4.1, our results on this model suggest that there may be still other factors that contribute to the failure of SSMs at extrapolating on state tracking to longer sequences, and could potentially point to other possible research directions.
>
> **Comment.**  Terminology inconsistency: the selective copy task is sometimes referred to as "denoising". Using consistent terminology would improve clarity.
>
> **Answer.** We thank the reviewer for finding this inconsistency and have fixed it.
>
> -----
>
> **References**
>
> Grazzi et al., Unlocking State-Tracking in Linear RNNs Through Negative Eigenvalues, International Conference on Learning Representations, 2025.
>
> Merrill et al., The Illusion of State in State-Space Models, International Conference on Machine Learning, 2024

---

### Review · Reviewer_pQWm · 2025-03-31

**Summary Of Contributions:**

This work performs controlled (synthetic) experiments to examine the ability of different sequence modeling networks to learn target memories in varied tasks.

**Audience:**

Yes

**Broader Impact Concerns:**

There are no broader impact concerns on the ethical side.

**Claims And Evidence:**

Yes

**Requested Changes:**

See the "Weaknesses" section.

**Strengths And Weaknesses:**

Strengths:
1. The paper is well-written and easy to follow.
2. Extensive controlled experiments are conducted.

Weaknesses:
1. There are only enumerations of experimental observations on the final performance. This work would be strengthened by more quantitative (at least, principled) fine-grained analysis of similarities/discrepancies among models/tasks.
2. It is not clear to determine the boundary of experimental insights derived in this work. For real-world tasks in practice, how can we leverage these insights? Do these insights still hold?
3. How can we guarantee fair comparisons among models given varied tasks? For tasks where certain networks fail or do not have superior performance, is it due to models' intrinsic capacity or hyper-parameter tuning?

---

> ### Author Response · Authors · 2025-07-15
> **Author Response (1/2)**
>
> We sincerely appreciate the reviewers’ detailed feedback and valuable suggestions. We are pleased that they found our comprehensive comparison of memory and generalization in RNNs, SMMs, and Transformers to be a novel contribution (Jfrh, pQWm, nk7E) and our paper well-written (pQWm, nk7E). We have carefully considered the reviewers’ comments and concerns and have addressed them below.
>
> -----
>
> **Comment.** There are only enumerations of experimental observations on the final performance. This work would be strengthened by more quantitative (at least, principled) fine-grained analysis of similarities/discrepancies among models/tasks.
>
> **Answer.** We wholeheartedly agree that providing more intricate analysis of the specifics of models could improve the observations from our experiments. We refined our analysis and made more direct relationships between the results and architecture specificities. Some of those additions appear at the end of each Section 4.1, 4.2 and 4.3. Specific to the counting task, we also included an analytical analysis that sheds some light on how a simple SSM model like S4D could have performed better on our counting task compared to a more sophisticated SSM model like Mamba.
>
> **Comment.**  It is not clear to determine the boundary of experimental insights derived in this work. For real-world tasks in practice, how can we leverage these insights? Do these insights still hold?
>
> **Answer.** This raises an important point. Although these tasks appear simple in isolation, they are often considered indicative of a model’s ability to perform on real-world problems. The idea is that by identifying a model's deficiencies on these benchmarks, improving its performance on them could lead to gains on related real-world tasks. However, we believe that these presumed correspondence between synthetic and real-world problems require further investigation to be validated.
>
> An illuminating example is offered by [Grazzi et al., 2025], who demonstrate that adjusting Mamba’s eigenvalues to solve the basic state tracking task of parity yields notable performance improvements on tasks such as MathHard and CodeParrot. These tasks are thought to rely on effective state tracking. Nonetheless, we acknowledge that such findings require broader verification across a wider range of tasks before drawing general conclusions.
>
> Some of these details are provided in our taxonomy, which we will improve to better clarify this relationship with real-world tasks.

---

> ### Author Response · Authors · 2025-07-15
> **Author Response (2/2)**
>
> **Comment.** How can we guarantee fair comparisons among models given varied tasks? For tasks where certain networks fail or do not have superior performance, is it due to models' intrinsic capacity or hyper-parameter tuning?
>
> **Answer.** While we can always attribute some part of the failures to the learning dynamics and the optimization method, for some cases we know that it is highly probable that intrinsic limitations are hindering the model's performance.
>
> The most notable case among our experiments is state tracking. As we have mentioned in the paper, [Grazzi et al., 2025 and Sarrof et al., 2024] have theoretically proved that SSM models in finite precision (including Mamba and S4D) are unable to learn the correct algorithm to solve state tracking tasks such as parity, modular counting (which is very similar to the counting task in our paper) and Z60, A4, A5, which are even more difficult; the reason being that their transition matrix needs to be input-dependent for solving these tasks (which is the case only for Mamba and IDS4) and also this matrix should include complex eigenvalues (which is not the case for Mamba). Therefore, we can expect that these models do not have sufficient capacity/expressivity to learn solutions similar to those learned by non-linear models like RNNs (for IDS4, though, we still don't have an explanation. See our point 2 in our summary of results at the end of section 4.1). For Transformers also, there is an extensive literature that explains how attention-based models are expected to fail on sensitive languages, like parity (sensitive in the sense that the output depends on every single token in the input, which is true for all our state-tracking tasks as well). For this, please refer to [Hahn, 2020].
>
> On the other hand, for counting experiments, we do not expect the failure of models in generalizing to be solely due to intrinsic limitations. For example, we see that RNN is able to solve and generalize in all setups, while GRU or LSTM or NRU sometimes fail. In theory, these models all have more capacity than RNN, so we believe that their failure is due to the training dynamics that somehow ends up with an incorrect solution for the task. It can also be the case that the excessive capacity of these models makes them overfit to our simple counting task. This is our hypothesis, but we have not conducted any careful analysis of the solutions learned by gradient descent. However, this is definitely a very important question to address and we plan to investigate it in our follow-up work. Regarding the failure of Transformer and Mamba, we hypothesise that it is intrinsic though. In the paper, we suggest a solution that S4D can learn (See the new Appendix, I), but is out of reach of Mamba, due to the fact that S4D has its transition matrix parameterized in complex space, while for Mamba, this matrix is real, and even limited to be non-negative-valued. For Transformer, we think it is because of the lack of positional encoding.
>
> Finally, regarding the copying task, we believe that this also is due to some intrinsic issues, as already brought up in [Deletang et al., 2023], where based on very similar observations (for them the task was double-string in formal language) they conclude that only recurrent networks augmented by some external memory (what they call Tape-RNN) can extrapolate on this task, due to it residing higher in the Chomsky hierarchy. They also interestingly conclude that scaling up the model, for example by adding more layers, cannot “help the model to climb up the hierarchy” as they interpret it.
>
> We have extended our analysis at the end of each experimental Sections 4.1, 4.2 and 4.3 to discuss these points in the paper as well.
>
> **Comment.** Requested Changes: See the "Weaknesses" section.
>
> **Answer.** We thank the reviewer for the useful comments. We answered the weaknesses in the revised paper. (See above)
>
> -----
>
> **References**
>
> Deletang et al., Neural Networks and the Chomsky Hierarchy, International Conference on Learning Representations, 2023.
>
> Grazzi et al., Unlocking State-Tracking in Linear RNNs Through Negative Eigenvalues, International Conference on Learning Representations, 2025.
>
> Michael Hahn, Theoretical Limitations of Self-Attention in Neural Sequence Models, Transactions of the Association of Computational Linguistics, 2020.
>
> Sarrof et al., The Expressive Capacity of State Space Models: A Formal Language Perspective, Neural Information Processing Systems, 2024.

---

### Review · Reviewer_Jfrh · 2025-07-01

**Summary Of Contributions:**

The paper studies the memory and generalization capabilities of RNN, Trasnformers, and SSMs. The paper provides empirical evidence on toy tasks such as copy, state tracking, and counting. The evidence shows that linear models may struggle to generalize to longer sequences compared to non-linear models.

**Audience:**

Yes

**Broader Impact Concerns:**

-

**Claims And Evidence:**

Yes

**Requested Changes:**

My main concerns are about the length of the paper and additional experiments, both of which are actionable as explained in the weaknesses section of my review.

**Strengths And Weaknesses:**

Strengths:
- The topic of memory and generalization is at the core of the design of recurrent models. In general, I like the experimental setup of the paper.
- While there are similar results scattered in the literature, I'm not aware of any other work providing a full comparison between RNNs and SSMs for memory tasks as done in this paper.

Weaknesses:
- the paper feels longer than necessary, and it would be strenghtened by making it shorter. For example:
	- it is unclear to me what is the value of having all the learning curves in the main text. Moving most of them in the appendix would increase the readability.
	- In general, the style is very verbose. It would be helpful to provide a table/figure that summarizes the benchmarks and results.
- given the empirical nature of the work, I expected a deeper analysis of the results. The results are a bit too minimal, especially considering the length of the paper. For example:
	- are the failures due to the model scale? what happens if we increase the size of the networks? Most models used in the paper seem very small, which may hinder the training dynamics
	- are there any hypothesis on why SSMs are not better?

Questions:
- "We define a memory taxonomy that applies to the deep learning synthetic tasks in our study, while relating those tasks and their memory requirements to corresponding tasks in formal language theory.": there are several studies in the area that study connections between formal languages and RNNs. I'm not sure that this contribution can be really considered as novel.

---

> ### Author Response · Authors · 2025-07-15
>
> We sincerely appreciate the reviewers’ detailed feedback and valuable suggestions. We are pleased that they found our comprehensive comparison of memory and generalization in RNNs, SMMs, and Transformers to be a novel contribution (Jfrh, pQWm, nk7E) and our paper well-written (pQWm, nk7E). We have carefully considered the reviewers’ comments and concerns and have addressed them below.
>
> -----
>
> **Comment.**  the paper feels longer than necessary, and it would be strengthened by making it shorter.
>
> **Answer.**  We agree that some parts of the paper can be better condensed, which we provide in an updated version of our submission. In particular
> We removed redundancies between the introduction and the related works section.
> We moved some details of the descriptions into the Appendix.
> We narrowed down some of the verbose descriptions and results to better focus on the provided tables and figures.
> We moved some figures that demonstrate redundancies in the observations to the Appendix.
>
> **Comment.** Are the failures due to the model scale? What happens if we increase the size of the networks? Most models used in the paper seem very small, which may hinder the training dynamics.
>
> **Answer.** We believe that the failures are not due to scale. The tasks of interest are generally not complex and do not require scale. For instance, [Merrill et al. (2024)] demonstrated that solvable state-tracking can be solved with one-layer models. For counting, our smaller RNN can learn the task and generalize. In our experiments, we selected the hyperparameters, including model size, to be sufficient and made sure they aligned with previous studies. Nonetheless, scaling would be an interesting future direction. We would like to refer the reviewer to our answer to the weakness 3 from the Reviewer pQWm. There, we discuss how, for some failure cases, we already know that it is due to intrinsic model limitations that regardless of the scale, impede models’ performance at long sequence lengths.
>
> **Comment.** Are there any hypotheses on why SSMs are not better?
>
> **Answer.** Several possible reasons have been proposed in previous studies. For example, [Merrill et al. (2024)] stated that SSMs cannot solve state-tracking problems due to a lack of input-dependent transition matrices (the A matrices in the underlying state-space model), while other works such as [Jelassi et al. (2024)] stated that SSMs struggle at the copying/denoising task simply due to a limited hidden state.
>
> Our analysis is complementary to these prior works by exploring SSM capabilities beyond solving tasks, notably how well they extrapolate. We believe the reason for their limited extrapolation capabilities compared to other models differs between tasks. For instance, on state-tracking, SSMs seem to learn an incorrect underlying transition matrix, due to the specific design choices of current SSMs as uncovered in the very recent work by [Grazzi et al.].  However, this hypothesis doesn’t apply to other tasks like counting. We also added an analysis that provides some evidence on how on our simple counting task, a simple SSM such as S4D  could  perform better than Mamba, which is in general a more powerful SSM model.
>  We have added our hypotheses on the reasons for the low generalization performance of SSMs  for each of the tasks in the appropriate Sections of the paper,  4.1, 4.2 and 4.3.
>
> **Comment.** Questions:  there are several studies in the area that study connections between formal languages and RNNs. I'm not sure that this contribution can be really considered as a novel.
>
> **Answer.** We definitely agree that there has already been work on studying formal languages and RNNs. We would like to clarify that our contribution is not in understanding RNNs (and other sequence models) from the perspective of formal languages, but rather attempting to do a similar study with a focus on the tasks stemming from deep learning literature rather than formal language. For instance, our copying task, which was suggested to benchmark different sequential models in terms of their ability to capture long-range dependencies, is similar to the formal language task of duplicate-string in [Deletang et al., 2023]. But we are not aware of other works analyzing such tasks with origins in deep learning from this point of view.
>
> **Comment.** Requested Changes: My main concerns are about the length of the paper and additional experiments, both of which are actionable as explained in the weaknesses section of my review.
>
> **Answer.** We have significantly shortened the paper and addressed the weaknesses in the revised manuscript.
>
> -----
>
> **References**
>
> Deletang et al., Neural Networks and the Chomsky Hierarchy, ICML, 2023.
>
> Merrill et al., The Illusion of State in State-Space Models, ICML, 2024
>
> Jelassi et al., Repeat After Me: Transformers are Better than State Space Models at Copying, ICML, 2024
>
> Grazzi et al., Unlocking State-Tracking in Linear RNNs Through Negative Eigenvalues, ICLR, 2025.

---

### Decision · Action_Editor_wGY8 · 2025-08-29

**Recommendation:** Reject

**Audience:**

Yes

**Audience Explanation:**

The topic addressed by the paper is timely. Many ML researchers are interested in learning in a sequential domain. The proposed comparison, i.e. linear vs non-linear models is interesting in view of the resurgence of interest in structured state-space models.

**Claims And Evidence:**

No

**Claims Explanation:**

Although all reviewers agree on the relevance of the proposed comparison, and that the authors rebuttal addressed most of their concerns, the claims on the results cannot still be considered fully supported by the experimental assessment.
There are concerns about the scope of the considered architectures, including the lack of a proper extended and systematic model selection process. In addition,  in the current version of the paper, there is no assurance that the observed behaviours will replicate on related real-world datasets. Finally, the paper could benefit from a further reduction in length.
Authors are invited to consider a resubmission addressing all the above concerns.

**Resubmission Of Major Revision:**

The authors may consider submitting a major revision at a later time.